



# Classification of synoptic circulation patterns with a two-stage clustering algorithm using the modified structural similarity index metric (SSIM)

Kristina Winderlich, Clementine Dalelane, Andreas Walter

Climate and Environment Consultancy Department, Deutscher Wetterdienst (German Meteorological Service), Offenbach am Main, 63067, Germany

*Correspondence to*: Kristina Winderlich (kristina.winderlich@dwd.de)

**Abstract.** We develop a new classification method for synoptic circulation patterns with the aim to extend the evaluation routine for climate simulations. This classification is applicable to any region of the globe of any size given the reference data.

Its unique novelty is the use of the modified structural similarity index metric (SSIM) instead of traditional distance metrics for cluster building. This classification method combines two classical clustering algorithms used iteratively, hierarchical agglomerative clustering (HAC) and k-medoids, with only one pre-set parameter - the threshold on the similarity between two synoptic patterns expressed as the structural similarity index measure SSIM. This threshold is set by the user to imitate the human perception of the similarity between two images (similar structure, luminance and contrast), whereby the number of

final classes is defined automatically.

We apply the SSIM-based classification method to the geopotential height at the pressure-level of 500hPa from the reanalysis data ERA-Interim 1979-2018 and demonstrate that the built classes are 1) consistent to the changes in the input parameter, 2) well separated, 3) spatially stable, 4) temporally stable, and 5) physically meaningful.

We demonstrate an exemplary application of the synoptic circulation classes obtained with the new classification method for

evaluating CMIP6 historical climate simulations and an alternative reanalysis (for comparison purposes): output fields of CMIP6 simulations (and of the alternative reanalysis) are assigned to the classes and a quality index is computed for the match in frequency and duration probability of these classes. We propose using this quality index to supplement a set of commonly used metrics for model evaluation.





# 1 Introduction

Research institutions around the world conduct climate studies and share their knowledge with the society and policy makers
through The Intergovernmental Panel on Climate Change (IPCC, www.ipcc.ch). The climate simulations used in the IPCC
reports are available to other scientists, besides those who run the models, through the Coupled Model Intercomparison Project
(CMIP, www.wcrp-climate.org/wgcm-cmip). The first two phases (CMIP1 and CMIP2) of this initiative addressed the ability
of numerical climate models to simulate the present climate and to respond to an increase of carbon dioxide concentration in
the atmosphere (Meehl et al., 1997; Meehl et al., 2000). The extended follow-up phase CMIP3 (Meehl et al., 2007) provided
output of coupled ocean-atmosphere model simulations of 20th-22nd century climate for the 4th Assessment Report (AR4) of
IPCC (www.ipcc.ch/report/ar4/syr/). As the number of climate simulations in the subsequent projects CMIP5 (Taylor et al.,
2012) and CMIP6 (Eyring et al., 2016) continued to increase, new requirements on the "quality" and "reliability" of such
simulations emerged. Having multiple models at their disposition, final users have a choice to use all models or only those
models, which pass a quality check i.e. an evaluation routine. Although testing and comparing models may create an illusion
of finding the best one in all its features, we emphasize here: there is no universally valid and absolutely objective evaluation
procedure for all purposes. It is important to include a broad suite of metrics into the evaluation spectrum, but various
applications may require different subsets of these metrics.

Hannachi et al (2017) emphasized the importance of the correct representation of weather regimes, their spatial patterns, and
persistence properties in global circulation models as they could properly simulate the climate variability and long-term
climatic changes under an external forcing such as, for example, the global warming. However, traditional techniques for
climate model evaluation, which are rooted in evaluation techniques for numerical weather prediction models, mainly focus
on individual variables and derived indices as summarized by Gleckler et al. (2008). These techniques use scalar variables,
called "metrics", and often illustrate symptoms of problems without explaining their causes that possibly may originate from
incorrect simulation of synoptic weather. As some studies have already demonstrated that the performance of a model varies
as a function of weather types (Díaz-Esteban et al., 2020; Nigro et al., 2011; Perez et al., 2014; Radić and Clarke, 2011) we
suggest to account for models synoptic behaviour in evaluation routines. But how to capture the correctness of the large scale
atmospheric dynamics in models?

The atmospheric circulation is a continuum that gradually changes and its dynamics can be described by a finite number of
representative "states"/"typical patterns" i.e. classes. Hochman et al. (2021) showed that such representation of the atmosphere
by quasi-stationary circulation patterns, often also termed as weather regimes, is a physically meaningful way to describe the
atmosphere (and not only a useful statistical categorization as it may be argued). Muñoz et al. (2017) also suggested using the
weather-typing approach to diagnose a range of variables in a physically consistent way helping to understand causes of model
biases. For evaluation purposes, any climate model simulation can be represented as a sequence of typical synoptic situations,
previously classified. Common variables used for representing the synoptic circulation are the sea level pressure, geopotential





heights and wind vector fields. Statistical measures, such as frequency and duration of each class, computed from the assigned
      sequence can be evaluated against reference data derived, for example, from a reanalysis.

      Many questions arise when building a classification of weather situations:

      -    On which spatial and temporal scales should weather situations be classified?

      -    Do the frequency and persistence of each weather situation play a role in the classification?

-    How many classes are sufficient to describe the atmospheric circulation?

      Answers to these questions are not trivial and strongly depend on the purpose of the classification.

      Weather situations are often described as patterns of positive and negative anomalies of geopotential (Hochman et al., 2021;
      Fabiano et al., 2020; Fettweis et al., 2010) or surface pressure (Lund, 1963; Beck et al., 2007), or a combination of both
      (Cannon, 2012; James, 2006) seen together at the horizontal scale of about 1000 km (synoptic scale). Weather patterns can

be defined at a regular temporal step, typically one day (Lamb, 1972; Hess and Brezowsky, 1952; Fabiano et al., 2020;
      Cannon, 2012) and be classified independent on their duration (James, 2006; Cannon, 2012; Beck et al., 2007; Fettweis et
      al., 2010). Alternatively, only recurrent, quasi-stationary and temporally persistent states of the atmospheric circulation
      would be classified (Dorrington and Strommen, 2020; Hochman et al., 2021) eliminating short-term patterns in the final set
      of classes.

There is no "universally correct" recipe on how to build synoptic classes and how many of them. Each application requires a
      number of classes constructed in a way best suitable for its purposes. A set of classes can be determined subjectively by an
      expert, as the well-known Hess-Brezowski Grosswetterlagen (Gerstengarbe and Werner, 1993; James, 2006; Hess and
      Brezowsky, 1952) or the Lamb weather types (Lamb, 1972), or using an automated classification method. Multiple different
      synoptic classifications have been developed over years as summarized by Yarnal et al. (2001) and Huth et al. (2008). An

overview and systematization of existing classification methods for synoptic patterns was compiled in a joint effort of
      multiple European Institutions in the COST Action 733 and summarized in the final project report (Tveito et al., 2016). A
      large number of classes is often used in classification methods that root in synoptic meteorology. Such methods - for
      example, the ZAMG-classification with 43 classes (Baur, 1948; Lauscher, 1985) and the Grosswetterlagen-based
      classification by James (2006) with 58 weather types (29 for winter and 29 for summer) - give priority to a high structural

differentiation among synoptic patterns, at the same time trying to maximize the homogeneity inside classes. This attempt
      may produce some classes, which have a small number of members or could be even empty.  On the other hand, methods
      that use a small number of classes focus on large-scale circulation regimes and can be used for investigating possible
      precursors for their changes, for example shifts of the jet stream (Dorrington and Strommen, 2020). These methods may
      handle the pattern diversity in a sub-optimal way: prioritize a low number of classes over the high intra-class homogeneity

and leave multiple synoptic patterns unclassified.

      Our purpose is to extend a traditional evaluation routine for climate models, which typically rests on a set of metrics for scalar
      variables (Gleckler et al., 2008), by a set of diagnostics considering the correctness of weather pattern representation. We are
      not the first ones to evaluate model dynamics in such way. Riediger and Gratzki (2014) evaluated climate indices (mean values



or hot, cold, wet and dry days) computed for five global circulation models and a reanalysis conditioned on different weather
types in a recent and a future climate using a threshold based classification method for the Central European region (Dittmann
et al., 1995). Cannon (2020) used two atmospheric classifications constructed on two reanalyses for evaluating historical
simulations of 15 pairs of global climate models from CMIP5 and CMIP6 datasets; the number of circulation classes used in
this study was 16 as suggested in the COST733cat database over smaller European domains (Philipp et al., 2010). Herrera-
Lormendez et al. (2021) used Jenkinson-Collison classification adapted to Europe for evaluating some CMIP6 models against
three reanalyses and analysed future changes in circulation for these models.

A "good" set of weather types should be able to describe all physically admissible states and events in the climate system i.e.
rain fall events and heat periods can be explained by an occurrence of individual weather types or a particular sequence of
certain weather types. Muños et al. (2017) and Nguyen-Le and Yamada (2019) investigated rainfall intensities in dependence
of weather types. Adams et al. (2020) found that extreme temperature events, as well as cold anomalies, are related to
circulation patterns. As we keep in mind the possible linkage between weather types and extreme weather, we would like to
have an automated classification that

-    produces structurally differentiated classes (similar as it is done in synoptic meteorology),
-    is applicable to any domain on the globe,
-    provides high homogeneity inside classes, and
-    encompasses almost all synoptic situations leaving no/or very few situations unclassified.

In our opinion, the last condition is especially important because rare synoptic situations may be linked to severe weather and
should be carefully handled in the evaluation procedure for climate models. Therefore, the wide variety of classification
methods that focus on very few quasi-stationary weather regimes do not suite our purpose as they eliminate rare synoptic
patterns from the analysis. Semi-automated classifications do not suite our purpose of model evaluation either, because these
methods require expert knowledge to define weather types in the considered region; this would limit our future options of
evaluation to only regions with available experts' knowledge.

The majority of classification methods included in the COST733cat database (Philipp et al., 2010) and in the literature (Cannon
et al., 2001; Hochman et al., 2021; Grams et al., 2017; Muñoz et al., 2017; Fabiano et al., 2020) uses the *k-means* clustering
algorithm (Milligan, 1985) in conjunction with the Euclidean distance as a metric to measure the degree of similarity between
clustered data elements. In this paper we elaborate on the draw-backs of using mean fields as cluster centres in the classification
of atmospheric data fields and suggest an alternative representation of cluster centres.

Distance metrics typically used in classification algorithms are often Euclidean norms ($L^2$–norms), mean squared error (MSE)
or Pearson's Distance. MSE remains the standard criterion for comparing modelled and observed signals in climate science
and in optimization routines despite of its weak performance and serious shortcomings in comparing structured signals
(pressure or geopotential fields can also be seen as "images") as thoroughly discussed by Wang and Bovik (2009). Following
the suggestions of Wang and Bovik (2009), we refrain from using MSE as a distance measure in classifying weather types,



and propose using the alternative Structural Similarity Index Metric (SSIM) introduced by Wang et al. (2004) for comparing geopotential fields.

Following the above mentioned arguments, we introduce the new two-stage classification method for synoptic circulation
patterns as an alternative to existing methods of clustering. The novel approach allows accounting for rare synoptic situations, which may be linked to severe weather and builds synoptic classes automatically without prior experts' knowledge. This alternative method, in our opinion, bears its own scientific value, because as the very least it corroborates previous results, but it even improves upon those previous results in both statistical (number of classes is defined automatically) and climatological aspects (all synoptic situations are classified, applicable to arbitrary regions of the globe without further experts' knowledge).

The novelty of this method consists of the following features:

-   it classifies all input data without pre-filtering and pre-initialization of classes,
-   it builds classes with strong structural differentiation and high inter-class homogeneity,
-   it uses a structural similarity metric instead of a distance-metric for classifying data,
-   it represents classes by their medoids instead of centroids, and
-   it uses an iterative combination of the hierarchical agglomerative algorithm with a partitioning *k-medoids* algorithm to determine the number of clusters automatically.

This classification algorithm does not need an initial distribution of elements and gradually continues building and reviewing clusters until there is no more clusters to be built and reviewed according to a given threshold of similarity.

We demonstrate that the new classification produces a set of well-separated classes, not necessarily of similar size, consistent
(small changes in the pre-set parameter do not alter classes strongly), stable with respect to the temporal selection of data (for example randomly chosen and shuffled), stable across various spatial resolutions and data volumes, and physically interpretable (i.e. final classes represent real synoptic situations).

In this paper we describe the new classification method and demonstrate its application to evaluation of global circulation models. The final result of the evaluation is expressed as a quality index that can be computed on models statistics.

The paper is structured in the following way: 1) introduction, 2) data and domain description, 3) description of the classification method, 4) presentation of resulting classes and 5) use of the derived classes in computing the quality index for evaluating CMIP6 climate simulations, 6) our conclusions and an outlook for future applications.

## 2 Data

We use four datasets in this study.

The first dataset, a dataset of synthetic data, is used to demonstrate the performance of the classification method explaining why modifications to the classical *k-means* algorithm are necessary. We generated this synthetic data set using Gaussian-shaped anomalies trying to mimic the smooth shape of geopotential patterns (the real data we wish to use later) and to illustrate how such anomalies are treated by the classification algorithm. The original circular shapes of the synthetic generated data



help to illustrate how such shapes are grouped into classes by classifications in a simpler manner as if we would have used the

real data for this demonstration.

The synthetic data of 1000 elements was generated for the domain of 22x22 grid points. Each field includes one large and 10 smaller Gaussian-shaped superimposed anomalies with randomly chosen sizes, randomly placed within the domain and with randomly chosen signs (negative or positive anomaly); additionally, a randomly generated linear shift of the mean is added to each field. Examples of generated anomaly fields are shown in Figure 1.


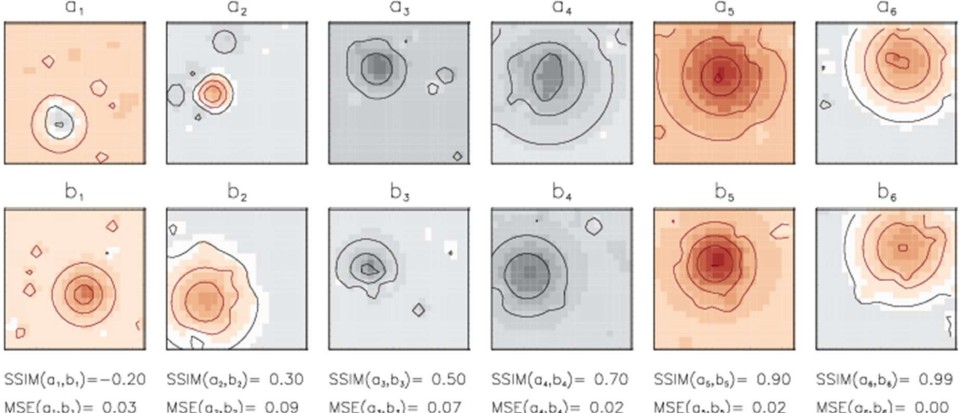

**Figure 1: Examples of synthetic data fields. Fields are shown pairwise ($a_1$-$b_1$, $a_2$-$b_2$,..) for demonstrating how visually perceived similarity is quantified in terms of SSIM and MSE given under the lower plot for each pair. Contour lines show the amplitude of negative anomalies (black) and positive anomalies (red) with interval of 0.25. Pairs are ordered by their SSIM-values from a**

**dissimilar pair (on the left) to a strongly similar pair (on the right). Note: smaller MSE does not guarantee larger SSIM-values as for the pair $a_1$-$b_1$.**

The second dataset, reanalysis data, the Reanalysis ERA-Interim (Dee et al., 2011) for the period of 1979-2018 is used as a realistic historical representation of the atmospheric circulation in Europe. The original spatial resolution of this data is approximately 80 km (T255 spectral) on 60 levels. Simulated synoptic regimes are represented by the geopotential height ($zg$)

at the pressure level of 500hPa sampled daily at 12:00 UTC for two practical reasons: 1) it often matches the mid-day peak in extreme weather conditions and 2) it is a typically available time for model output (for subsequent model evaluation). Spatially $zg$ fields are sampled on a grid of 2°x3° as suggested by participants of the COST Action 733 (Tveito et al., 2016). The coarse-scale sampling is sufficient due to the fact that the synoptic-scale 500-hPa geopotential height does not require high resolution to reproduce the key physical mechanisms (Muñoz et al., 2017). The chosen domain (Figure 2) has 22x22 grid points with the

lower left corner at (20°W, 29°N).



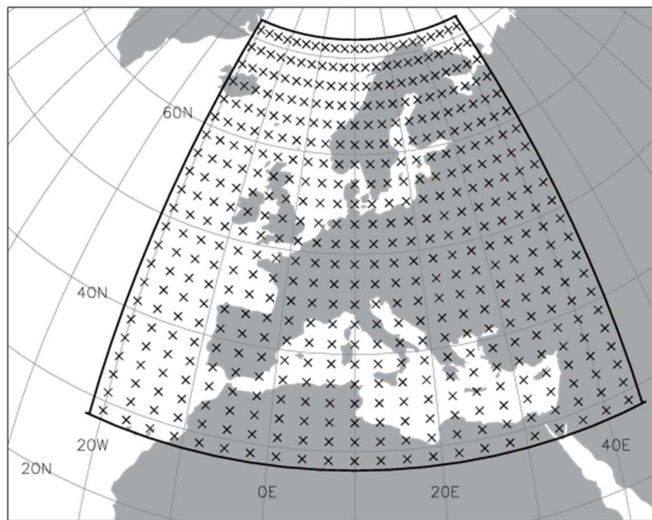

**Figure 2: Domain for classification of synoptic circulation patterns: crosses show sample points every 2° in latitude and every 3° in longitude directions, 22x22 grid points in total. The solid black line shows the outer edge of the domain.**

Some typical synoptic patterns may occur in different seasons but should be grouped into one class. To allow this, we pre-process the original geopotential height fields ($zg$): remove the seasonal amplitude from the original daily data and normalize the resulting fields by the daily standard deviation as in Eq. (1):

$$zg_a = \left(zg - \mu_{zg}\right)/\sigma_{zg} \tag{1}$$

The mean $\mu_{zg}$ and the standard deviation $\sigma_{zg}$ are calculated for each grid point and for each day of the year from the 40-years

of reanalysis data; both fields are smoothed in time with 151-days running average for each grid cell of the domain. The long smoothing period was chosen with the purpose to produce smooth seasonal curves of the mean 500hPa-geopotential and its standard deviation. Using such smooth mean and standard deviation curves for the normalization of the geopotential fields (prior to clustering) we preserve much of the fields anomaly. The resulting geopotential anomaly fields $zg_a$ are used in the classification.

The third data set is the alternative reanalysis NCEP1 (Kalnay et al., 1996). Assuming that the alternative reanalysis captures the synoptic circulation of the reference data ERA-Interim better than any unconstrained global circulation model, the evaluation of an alternative reanalysis gives an estimate of the upper bound for attainable value of the quality index.

   The 4[th] dataset of climate model output from the Coupled Model Intercomparison Project Phase 6 (CMIP6, https://www.wcrp-climate.org/wgcm-cmip/wgcm-cmip6, (Eyring et al., 2016)). Based on data availability we chose 32 global circulation models

for the historical period 1979-2014, preferably simulation version r1i1p1f1, when available, or r1i1p1f2/r1i1p1f3 otherwise.



We use the output data for geopotential at 500hPa of 32 the chosen models to demonstrate a possible evaluation routine that uses the synoptic classes derived on the reference reanalysis.

## 3 Method

A frequently used approach for identifying circulation regimes is to apply the *k-means* clustering algorithm (Milligan, 1985) to the synoptic circulation data: an overview can be found in the COST733cat database (Philipp et al., 2010) and in the recent literature (Cannon et al., 2001; Hochman et al., 2021; Grams et al., 2017; Muñoz et al., 2017; Fabiano et al., 2020). *K* is the number of classes to be built (this number must be set prior to the classification) and *Means* denotes the average of data elements within each class (also called centroid). The *k-means* method partitions the input data into *K* clusters, so that each data element belongs to the cluster with the nearest centroid minimizing within-cluster variances; the *k-means* method is simple and always converges to a solution. Although *k-means* and its multiple variants are commonly applied in the field of the atmospheric science, they exhibit serious limitations with regard to our aims:

1) use centroids (means of all elements in a cluster) to represent classes: using mean fields as cluster centres in the classification of atmospheric data fields may be suboptimal and lead to building classes with dissimilar elements (as shown later in this paper),

2) require a pre-specified number of classes,

3) use structure-insensitive distance metrics (e.g. Euclidean distance) for the optimization of the element assignment among classes. The *k-means* clustering assigns every data element to the cluster centre that is closest to it. This makes the method sensitive to noise in the data and may lead to an assignment of a data element to a structurally dissimilar cluster centre (Falkena et al., 2021); a pair of data fields is structurally dissimilar when it shows patterns perceived by an observer (or characterized with any structural similarity measure) as dissimilar.

The mean squared error (MSE) and the Pearson correlation coefficient (PCC) are probably the dominant quantitative performance metrics in the field of model evaluation and optimization. The *k-means* clustering algorithm typically uses the MSE to measure the distance between clustered data elements. However, Wang and Bovik (2009) demonstrated that the MSE has serious disadvantages when applied on data with temporal and spatial dependencies and on data where the error is sensitive to the original signal. Mo et al. (2014) in turn demonstrated that the PCC as a metric is insensitive to differences in the mean and variance. However, atmospheric data (pressure, geopotential, temperature fields) often reveal dependencies in time and space, as well as shifts in the mean and differing variances. Both studies mentioned above (Mo et al., 2014; Wang and Bovik, 2009) recommend using an alternative measure for signal/image similarity, the Structural Similarity (SSIM) index, to quantify the goodness of match of two patterns. The SSIM (Wang et al., 2004) simulates the human visual system that "recognizes" structural patterns and error-signal dependencies, and shows a superior performance as a similarity measure over the MSE and PCC.





### 3.1 Structural Similarity Metric SSIM and its modification.

We use the Structural Similarity index SSIM (Wang et al., 2004) for measuring the similarity between synoptic patterns (SP) represented by the geopotential height anomalies $zg_a$. These fields are highly structured images, meaning that the sample points

of these images have strong spatial dependencies, and these dependencies carry important information about the structures of the highs and lows in the field. The SSIM incorporates three perception-based components of image difference: structure (covariance), luminance (mean) and contrast (variance):

$$SSIM(x,y) = \frac{(2\mu_x\mu_y+c_1)(2\sigma_{xy}+c_2)}{(\mu_x^2+\mu_y^2+c_1)(\sigma_x^2+\sigma_y^2+c_2)} \qquad (2)$$


where

$x, y$ – non-negative signals/images,

$\mu_x, \mu_y$ - average values for $x$ and $y$,

$\sigma_x, \sigma_y$ - variance for $x$ and $y$,

$\sigma_{xy}$ - covariance of $x$ and $y$,

$c_1, c_2$ - stabilizing constants for weak denominator.

For each pair of images the *SSIM-value* is computed, which ranges $-1 \leq SSIM(x,y) \leq 1$. The *SSIM(x,y)=1* if and only $x=y$ (x and y are two identical images). In practice, most *SSIM-values* are positive and *SSIM(x,y)<1* identifying some difference between two images. Negative values of SSIM only occur when the covariance term $\sigma_{xy}$ is negative. The *SSIM-value* is usually computed

for multiple sliding windows inside the image. But for simplicity here, only one *SSIM-value* is computed for the whole domain. As the selected domain is relatively large and extends to high latitudes, areal weighting was applied to all fields prior to computing SSIM.

From the formulation of SSIM (2) it is important to note that it is applicable as similarity metric only to same-sign data. However data in climate-related applications are often mixed-sign and/or normalized (with the mean around zero). Therefore,

SSIM in its original form (2) cannot be used as the product of means $\mu_x$ and $\mu_y$ with different signs in combination with the negative covariance term $\sigma_{xy}$ as it would yield a positive *SSIM-value*. To overcome this limitation Mo et al. (2014) proposed to "shift" x and y by the minimum value of the two fields: $x' = x - \psi_{xy}$ and $y' = y - \psi_{xy}$ are non-negative, where $\psi_{xy} = min(x_n, y_n| n=1, 2, ... ,N)$. However, this modification weakens the sensitivity of SSIM to the difference between the means as a result of enlarged denominator.

We suggest an alternative modification, which only moderately modifies the magnitude of the denominator preserving the difference between the original means $\mu_x$ and $\mu_y$:

$$SSIM(x,y) = \frac{(2\mu'_x\mu'_y+c_1)(2\sigma_{xy}+c_2)}{\left((\mu'_x)^2+(\mu'_y)^2+c_1\right)(\sigma_x^2+\sigma_y^2+c_2)} \qquad (3)$$





where

$\mu'_x = \frac{|\mu_x|+|\mu_y|}{2}$ (4)

$\mu'_y = \mu'_x + |\mu_x - \mu_y|.$ (5)

The latter formulae (3) is applicable to floating-point data with mixed-sign.

The choice of stabilizing constants is "somewhat arbitrary" as SSIM is "fairly insensitive" to these values - the authors say (Wang et al., 2004). Baker et al. (2022) suggest that $c_1$ and $c_2$ should be the same as to make both terms equally influential

and propose $c_1$ and $c_2$ to be "small enough to not disproportionately influence" the final *SSIM-value*. We set $c_1 = c_2 = $ 1e-8 as suggested by Baker et al. (2022).

### 3.2 Examples of MSE and SSIM as a measures of similarity.

We get the first glance at the ability of SSIM to capture the structural similarity from its application on the synthetic data (Figure 1). Figure 1 shows fields of the synthetic pairs of maps $(a_i, b_i)$ for *i=1,2,..* and their respective values of structural

similarity *SSIM($a_i, b_i$)* and the mean square error *MSE($a_i, b_i$)* under each pair. Assuming MSE and SSIM are capable measures for similarity between two signals we expect MSE to decline with growing SSIM as the distance between more similar signals should be smaller than the distance between less similar signals. We ordered exemplary pairs in Figure 1 by their increasing *SSIM-value* (from left to right side of the plot), but see that respective *MSE-values* of these pairs do not decline monotonously with increasing similarity. It is remarkable that *MSE($a_1, b_1$)< MSE($a_2, b_2$)* i.e. the distance between the pair of signals $(a_1, b_1)$ is

smaller than the distance between the pair $(a_2, b_2)$, although signals $a_1$ and $b_1$ are obviously less similar to each other (they have anomalies of different sizes and placements) as signals $a_2$ and $b_2$. This example implies that using MSE in a clustering algorithm would rather group the pair $(a_1, b_1)$ into the same class than the pair $(a_2, b_2)$. Such preference results from the insensitivity of MSE to the spatial correlation and could lead to building classes with structurally dissimilar members. Using SSIM instead of MSE helps to capture the degree of similarity between clustered signals in a better way on the exemplary synthetic data.

Now we show that SSIM applied to real geopotential height anomalies $zg_a$ is preferable over MSE: in Figure 3 both pairs of geopotential anomalies have *MSE(a,b)=MSE(a,c)=0.1*, however the pair *a-b* has a high value of *SSIM(a,b)=0.7*, whereas for the other pair *SSIM(a,c)=-0.3* that indicates dissimilarity. In other words, the MSE does not detect "obviously" dissimilar geopotential anomaly patterns.


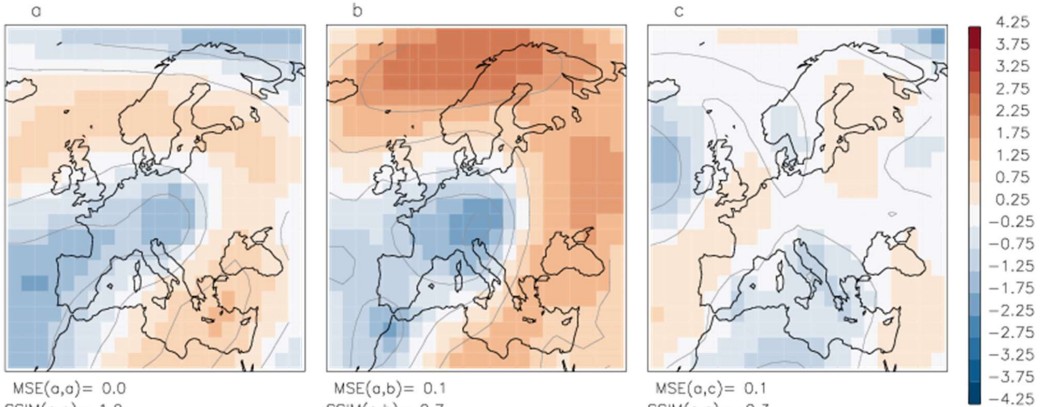

**Figure 3: The pairs of geopotential anomaly fields *a-b* and *a-c* have same small *MSE=0.1* but are strongly different in terms of SSIM: *SSIM (a, b) = 0.7* means fields *a* and *b* are similar, *SSIM (a, c) = -0.3* means fields *a* and *c* are dissimilar. Contour lines show the amplitude of anomalies with interval of 1.**

These two examples, with the synthetic data (Figure 1) and the real geopotential data (Figure 3), illustrate the weakness of the MSE as similarity metric for comparison (and subsequent clustering) of structured data fields as compared to the SSIM. Therefor we propose using SSIM as similarity measure in a new clustering algorithm.

### 3.3 Modifications of *k-means* applied to synthetic data

For supporting our previous arguments (about deficiencies of MSE distance measure) we setup and run three experimental

classifications on the synthetic data: 1) the classical *k-means* clustering algorithm with the distance measure MSE (*k-means-MSE*), 2) the *k-means* with the alternative similarity measure SSIM (*k-means-SSIM*), and 3) the *k-medoids* with the similarity measure SSIM (*k-medoids-SSIM*). For obtaining comparable results, we initialize all three experimental classifications with the same 9 class centres (Figure 4a), which we a-priori derived by an independent run of the hierarchical agglomeration clustering (*HAC*) algorithm (with the SSIM-measure for cluster merging) on the synthetic data set. The *HAC*-algorithm belongs

to a family of hierarchical algorithms and uses a completely different strategy for cluster building as opposed to the partitioning algorithms *k-means* and *k-medoids*. This insures that the classes obtained with *HAC* for the subsequent initialization of the *k-means* and *k-medoids* algorithms do not provide any hidden advantage for either of these algorithms.





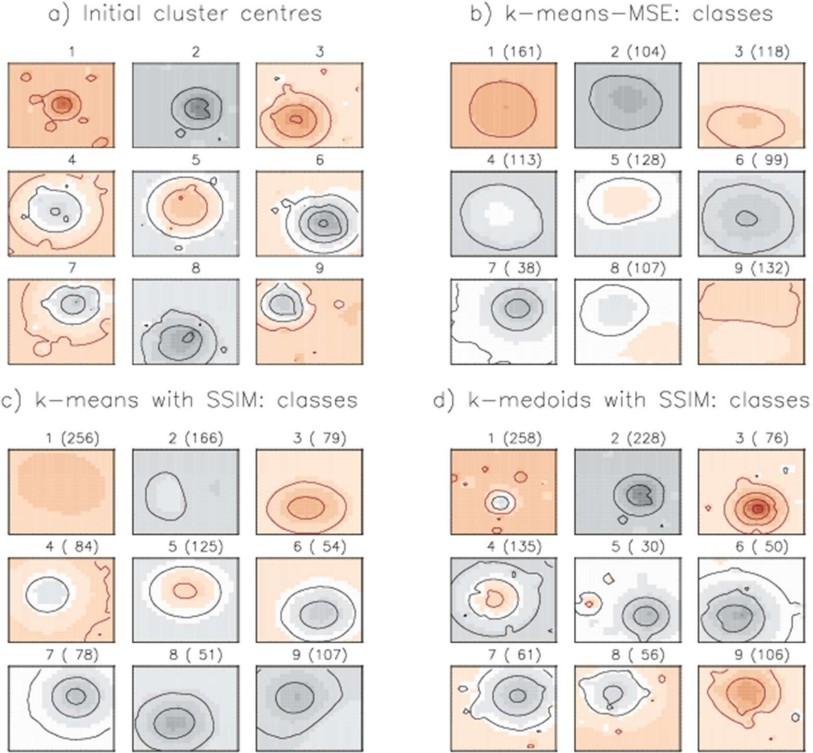

**Figure 4: Cluster centres: a) derived by *HAC*-algorithm and used for the subsequent initialization of the partitioning algorithms,**
**resulted from b) *k-means-MSE* classification, c) *k-means-SSIM* classification, and d) *k-medoids-SSIM* classification. Contour lines**
**show the amplitude of negative (black) and positive anomalies (red) with interval of 0.25. In panels b), c) and d) above each plot**
**numbers of elements (in brackets) in each class are shown.**

Results of *k-means-MSE* classification (Figure 4b): these class centres (centroids) visibly deviate from the corresponding

initialization fields (Figure 4a) in first place by the reduced magnitude of anomalies as a result of averaging multiple fields.

Classes 3, 5 and 9 also have skewed shapes of anomalies, originally circular, as a result of averaging multiple patterns with

variously placed anomalies. We already showed (Figure 1 and Figure 3) that small MSE does not guarantee the structural

similarity of compared patterns. Classes built with *k-means-MSE* show very little structural detail as a result of building cluster

centroids over multiple class elements, whose structural similarity remained unaccounted. The danger of having such classes

"with vanishing structure" is that they may serve as attractors for further elements as the clustering algorithm runs targeting at

minimizing MSE only. This leads to the so-called "snowballing" effect i.e. the more elements are assigned to this class, the

less structure shows its centroid, the more elements are assigned and so on. Cluster 9 (Figure 5) is a good example of such

"snowball"-class: although all shown elements have comparable small MSE to the final class centre, their visual (for an





observer) and computed similarity (value of SSIM) differs strongly as shown for a group of the first 28 elements (out of 132) indicating a strong structural inhomogeneity of patterns contained in one class. This example demonstrates the danger of

building "snowball" classes when using MSE as distance metric for data with highly structured patterns.

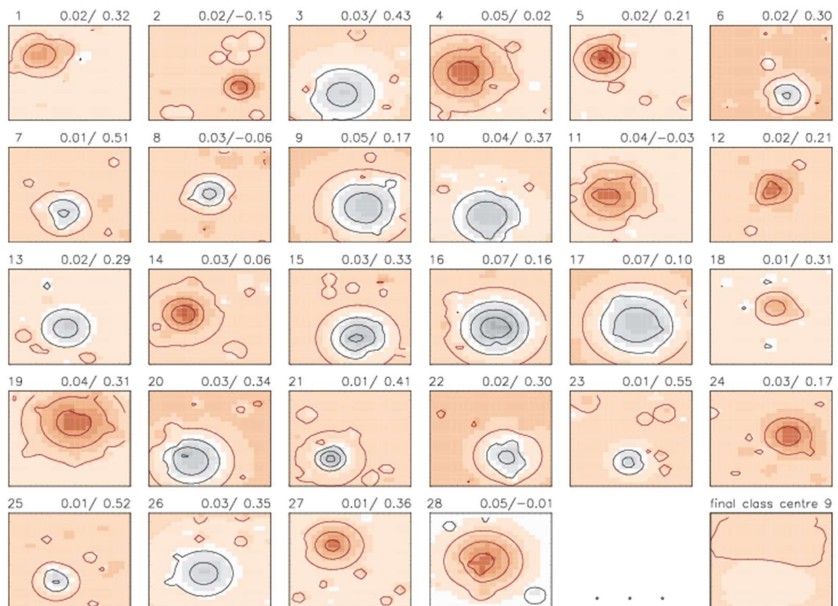

**Figure 5: An example of a "snowball"-class resulted from the classification *k-mean-MSE*: class centre 9 (bottom right plot) has very weak structure, whereas its first 28 elements (only shown as examples out of 132) have strong anomalies of mixed sign and placed in various locations of the domain. Values of MSE and SSIM for an element to the class centre 9 are shown above plots for each element**
**in form <MSE>/<SSIM>.**

Results of *k-means-SSIM* classification (Figure 4c): in an attempt to avoid building structurally inhomogeneous clusters we replaced the Euclidean distance metric MSE by the structural similarity measure SSIM in the classification algorithm, yielding the *k-means-SSIM* classification. Retrieved classes show some structural patterns that resemble the initial anomaly patterns in the data, although weakly pronounced: the amplitude of the large anomaly is reduced and smaller anomalies have nearly

vanished by averaging. We see: using SSIM instead of MSE helped to preserve circular shapes of the initial anomalies to some degree, implying that only structurally similar patterns are grouped into one class. However, resulting classes are too smooth in structure (reduced amplitude of anomalies) due to averaging by building centroids.

Assuming the synthetic data represents some physically meaningful field, for example a pressure or a geopotential field, the weakening of the anomalies amplitude by *k-means-SSIM* and *k-means-MSE* may have serious implications on the



interpretability of the resulting classes i.e. these classes do not represent any of the original data elements and, therefore, none of the realistic states of the atmosphere associated with these data. Additionally, such smooth fields would not be able to represent synoptic situations with extreme gradients that may be linked to extreme weather.

We construct a new *k-medoids-SSIM* classification (Figure 4d) as we keep the similarity metric SSIM (it showed advantages in structure-preserving as compared to MSE) but replace the representation of cluster centres in the clustering algorithm by

single "representative" elements – medoids (Kaufman and Rousseeuw, 1990). A medoid is the element of the class with the smallest dissimilarity to all other elements in this class. Each medoid itself is part of the data. Retrieved classes (Figure 4d) show strong anomaly amplitudes and not necessarily resemble their initialization fields, except classes 2 and 7 those medoids remained nearly "untouched" by the classification. The distribution of cluster elements in *k-medoids-SSIM* is done at each step of the algorithm by computing the medoid (= element with most mean similarity to all cluster elements) of each cluster. This

procedure is less sensitive to the addition of new elements to the cluster than the re-computation of centroids: new cluster elements do not necessarily modify the clusters medoid defined at the previous step of the algorithm. This  robustness of the *k-medoids* algorithm (Kaufman and Rousseeuw, 1990) with regard to outliers and noise helps to avoid "snowballing" in cluster building and explains the match of classes 2 and 7 to their initial fields i.e. the initialization of these classes was a "good guess" that proved to be robust throughout the *k-medoids-SSIM* algorithm (Note: initial fields ought not to be "good guesses" and to

remain preserved by the algorithm).

Following the arguments resulting from the application of the three classification algorithms on the synthetic data with structured patterns and structured errors, we propose using *k-medoids* algorithm with the similarity metric SSIM for classification of the real geopotential data as only this algorithm builds a set of classes that represent data elements and include only structurally similar elements.

**3.4 Initialization of classes**

There are multiple ways of defining the number of classes for the partitioning algorithm *k-medoids* (similarly to *k-means*) ranging from a random guess to the analysis of the data based on principal component analysis PCA, also known as empirical orthogonal functions, Huth (2000). Lee and Sheridan (2012) suggested the initialization of the clustering algorithm by selected PCAs. The reason for this statement was the common assumption that the first few modes returned by PCA are physically

interpretable and match the underlying signal in the data. However, Fulton and Hegerl (2021) tested this signal-extraction method and demonstrated that it has serious deficiencies when extracting multiple additive synthetic modes (false dipoles instead of monopoles, which may lead to serious misinterpretation of extracted modes). They also found that PCA tends to mix independent spatial regions into single modes. Huth and Beranová (2021) demonstrated that unrotated PCAs (still often used) result in patterns that are rather artefacts of the analysis than true modes of variability. Additionally, methods that apply

the PCA-filtering to input data, do not suite our purpose as these methods eliminate rare synoptic patterns from the analysis taking into account only a few PCAs with the largest Eigen-values and prevent building classes for rare synoptic situations. Guarded by the above mentioned ideas, we decided not to use the PCA-based initialization of the clustering algorithm. For the



initialization of *k-medoids* we suggest using another classical clustering algorithm - hierarchical agglomerative clustering (*HAC*). An example of *HAC*-retrieved initial classes we described in chapter "3.3 Modifications of *k-means* applied to synthetic

data" with the synthetic data: the *HAC*-algorithm builds classes whose centres are used to initialize the subsequent partitioning algorithm. Furthermore, we suggest using a combination of the two clustering algorithms – *HAC* and *k-medoids* – interactively i.e. merge similar clusters at the first step (HAC) and distribute all data elements to the new clusters at the second step. This two-stage algorithm stops, when no similar clusters are left to combine. The centres (medoids) of final clusters give the set of classes. We describe the new two-stage clustering algorithm below.

**3.5 New classification method: two-stage clustering algorithm**

Following the previous considerations, we made three essential decisions to modify the classic *k-means* algorithm in order to construct an algorithm better suitable (from our perspective) for building classes of synoptic patterns:

- Decision 1: use an alternative similarity measure
- Decision 2: use medoids to represent classes

- Decision 3: use a two-stage algorithm for the stepwise determination of the number of classes

The two-stage clustering algorithm combines two clustering methods - the hierarchical agglomerative clustering (*HAC*) and the *k-medoids* clustering - in such way that the output from the first is used as input into the second and vice versa. It inherits the strengths of both contributing algorithms.

Initially each data element represents its own cluster. Similarity between each pair of synoptic patterns is computed as structural

similarity index metric SSIM. The *HAC* is a very flexible clustering method that can use any distance or [dis]similarity measure as it allows different rules for aggregating data into clusters (Schubert and Rousseeuw, 2021). At each step, *HAC* determines the number of clusters and their medoids using a threshold on the *SSIM-value* for merging similar elements into one cluster. The merging threshold $TH_{merge}$ is set by the user and intuitively means the minimal human-perceived similarity of a pair of data elements to be included/merged into one cluster. *K-medoids* builds clusters (similarly to the widely-known method of *k-*

*means*) using the medoid-prototypes and an arbitrary [dis]similarity measure SSIM for cluster elements (D'urso and Massari, 2019; Schubert and Rousseeuw, 2021): it rearranges all data elements among medoid-prototypes (an operation that *HAC* cannot do) in order to maximize the within-cluster homogeneity. *K-medoids* in few iterations produces optimized clusters. The new medoids are computed and initialize the next step of *HAC* and so on.

At each iteration of the two-stage clustering, the two steps are done in the following way:

1. **The 1st Step: *HAC* (merge clusters):**

1.1. Clusters with sufficient similarity $SSIM > TH_{merge}$ are merged: clusters with higher similarity are merged prior to those with lower similarity (the similarity between two clusters is measured as the similarity between their medoid fields)

1.2. Temporary cluster medoids are recomputed

2. **The 2nd Step: *k-medoids* (recompose clusters):**





2.1.    Temporary cluster medoids from the first step are used to initialize the *k-medoids* clustering algorithm

2.2.    Each data element is assigned to the cluster with the most similar medoid

2.3.    Cluster medoids are recomputed

2.4.    *K-medoids* clustering is repeated until an optimum (for the given number of medoids!) distribution of all data elements is achieved

Both steps are repeated until there is no sufficiently similar pair of clusters left to be merged.

The presented classification method, as any other classification method, requires some pre-set parameters. The final number of clusters produced by the two-stage clustering algorithm depends on the threshold $TH_{merge}$ for merging elements into clusters and, eventually, on the amount of data to be clustered. Although the choice of $TH_{merge}$ is crucial, there is no statistical or analytical formula for computing this threshold, it can only be chosen subjectively by comparing pairs of synoptic patterns (SPs) and asking observers about their perception of similarity. Examples of "similar" synoptic patterns are shown in Figure 6. We analysed multiple pairs of SPs and, based on the personal perception of similarity (our own as well as of persons not involved in the development of this classification method), estimated that the threshold value $TH_{merge}$ must lie between 0.40 and 0.45 for recognizable similarity i.e. pairs with *SSIM-value* less than 0.40 being generally perceived as dissimilar. Figure 6 illustrates examples of similarity between three exemplary reference SPs and arbitrarily chosen SPs with *SSIM-values* of 0.60, 0.50, 0.45, 0.40, and -0.10 to each reference. SPs with *SSIM≥0.60* are "strongly similar" to the reference, SPs with the *0.40≤SSIM<0.60* are "similar", and with the *SSIM<0.40* - "weakly similar" to the reference. SPs with *SSIM<0* are "dissimilar" to the reference as, by definition of SSIM, the negative values of SSIM result from negative covariance of compared patterns.

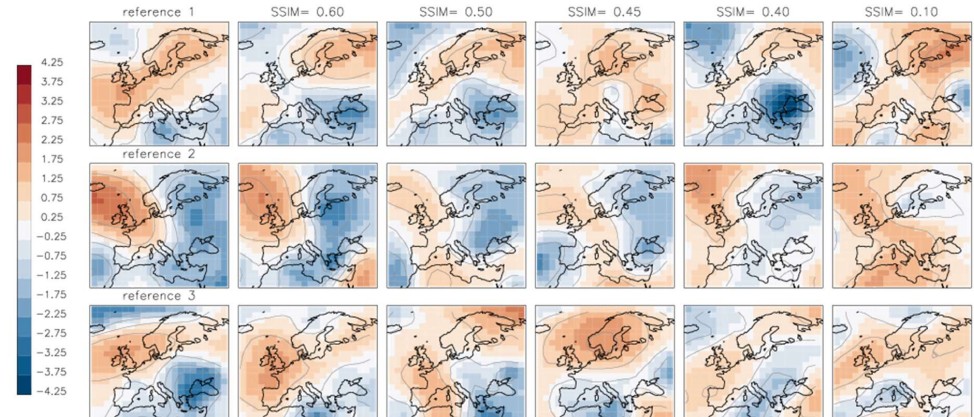

**Figure 6: Examples of three synoptic patterns $zg_a$ (left column "reference"). Each row contains examples of alternative synoptic patterns with the *SSIM-value* to the "reference". Contour lines show the amplitude of anomalies with interval of 1.**


The definition of the threshold $TH_{merge}$ implies that a reduction of its value loosens the requirement on data similarity for cluster

building and provides a smaller number of final classes. In contrary, an increase of $TH_{merge}$ tightens the requirement on the data similarity for cluster building and, therefore, leads to a larger number of final classes. At the same time, the higher $TH_{merge}$ also loosens the requirement of separation between classes and permits a higher similarity among them. Thus varying the value of $TH_{merge}$ may be used, to some extent, to steer the clustering algorithm to produce the number of final classes of a particularly desired magnitude.

Keeping in mind the intended application (evaluation of climate models) the question arises: how many classes do we need to describe the synoptic flow? In the present study, we use 40 years of daily synoptic patterns, 14600 daily data fields, which is a usual number of available reference data in climate research for the industrial time. How many classes do we need to represent synoptic situations of these 40 years? Would 10 or 100 be sufficient? The answer to this question is not trivial. The number of derived classes depends on the pre-set parameter $TH_{merge}$. Whereas, values of $TH_{merge}$ smaller than 0.40 were mainly discarded

by observers, testing higher values remains reasonable. We test three values for the threshold $TH_{merge}$ - 0.40, 0.425, and 0.45. Thus we produce three sets of classes whose separability in dependence of the $TH_{merge}$ can be analysed.

**3.6 Criteria for the evaluation of the clustering algorithm. Choice of the threshold $TH_{merge}$ for class merging.**

We analyse the performance of the new method using four criteria suggested by Huth (1996): The clusters should (i) be consistent when pre-set parameters are changed, (ii) be well separated both from each other and from the entire data set, (iii)

be stable in space and time, and (iv) reproduce realistic synoptic patterns.

Cluster consistency. The consistent evolution of classes implies that small changes in the pre-set parameter $TH_{merge}$ lead only to small changes in the classes. For illustrating the sensitivity of the clustering algorithm to the choice of $TH_{merge}$ it was run for three values chosen in the previous chapter, the reference value 0.40 and two higher values 0.425 and 0.45. Classes are consistent if an increase in the number of classes caused by a change in $TH_{merge}$ is realized predominantly by splitting a few

classes, with others remaining almost unchanged. Such evolution is difficult to quantify. The consistency of the clusters is illustrated by similarity diagrams - diagrams that resemble the "arrow diagrams" in Huth (1996) - for the sets of classes built with the varying parameter $TH_{merge}$.

Cluster separability. We calculate two metrics introduced in in the COST Action 733 report (Tveito et al., 2016) for characterizing the separability and within-class variability. Additionally we introduce a new indicator of class separability in

terms of similarity. The separation of clusters from randomly chosen data is addressed by the comparison of the metrics/indicators calculated on the clusters to the metrics calculated on "random groups". The "random groups" are generated for each cluster as groups of the same size but of randomly chosen data elements (one realization).

Metric 1: The explained variation $EV$ of the data is determined as the residual between 1.0 and the ratio of the sum of squares within classes (synoptic types) $WSS$ to the total sum of squares $TSS$:

$$EV = 1 - \frac{WSS}{TSS} \tag{6}$$


Metric 2: The distance ratio *DRATIO* is the ratio of the mean distance between elements assigned to the same class *DI* and the mean distance between elements assigned to different classes *DO*. The Euclidean distance is used for computing *DI* and *DO*:

$$DRATIO = \frac{DI}{DO} \tag{7}$$

We construct a new indicator *SSIMRATIO* for the class separability, similarly to the *DRATIO*, is defined as the ratio of the
mean similarity within classes (*SSIM$_{in}$*) to the mean similarity among different classes (*SSIM$_{out}$*):

$$SSIMRATIO = \frac{SSIM_{in}}{SSIM_{out}} \tag{8}$$

The mean similarity within classes *SSIM$_{in}$* is calculated as the mean "internal" similarity of all classes, where mean similarity-value of each element *j* to each element *k* of the same class *i* is computed:

$$SSIM_{in} = \frac{1}{n}\sum_{i=1}^{n} SSIM_{internal,i} \tag{9}$$

$$SSIM_{internal,i} = \frac{1}{m_i}\sum_{j=1}^{m_i}\frac{1}{m_i}\sum_{k=1}^{m_i} SSIM(j,k), \tag{10}$$

where *n* – number of classes, *m$_i$* – number of elements in class *i*, *SSIM(j, k)* – similarity of element *j* to element *k* of the same class *i*.

Mean similarity to other classes *SSIM$_{out}$* is calculated as the mean similarity of all class elements to all class elements of all other classes except its own:

$$SSIM_{out} = \frac{1}{n}\sum_{i=1}^{n} SSIM_{external,i} \tag{11}$$

$$SSIM_{external,i} = \frac{1}{m_i}\sum_{j=1}^{m_i}\frac{1}{\sum_{k=1,k\neq j}^{n} m_k}\sum_{k=1,k\neq j}^{n} SSIM(j,k), \tag{12}$$

where *n* – number of classes, *m$_i$* – number of elements in class *i*, *SSIM(j, k)* – similarity of element *j* to element *k* of any other class but not of the same class, $\sum_{k=1,k\neq j}^{n} m_k$ - number of all elements in all classes except class *j*.

Indicator *SSIMRATIO* could be viewed as an indicator of separability of classes in terms of pairwise similarity value: larger
values tell us about stronger within-class similarity in comparison to similarity of other classes.

**Note: After comparing the computed metrics and indicators, we discuss the choice of the threshold *TH$_{merge}$*. Once chosen, this value of *TH$_{merge}$* will be used for further analysis throughout the paper.**

According to the stop-criterion of the clustering algorithm, each pair of derived classes has similarity value less than *TH$_{merge}$* i.e. in the classification obtained with *TH$_{merge}$*=0.40 each pair of final class medoids is less similar to each other than this
threshold. Although the classes are represented by the cluster medoids in the clustering algorithm, it is also reasonable to require that the resulting cluster centroids (means) be at least not "strongly similar" (*SSIM<0.60*) to each other. We compute matrices of similarities for medoids and for centroids and analyse how well the medoid-separation algorithm provides the separation of centroids in the final set of classes.

Cluster temporal stability. The amount of input of synoptic data is crucial for building the representative set of classes. In
periods of only few years of data important synoptic circulations might be simply un- or under-represented because of long-term variability and, therefore, missing in the final set of classes. The clustering algorithm is run on a continuously increasing





data volume of 1,2,..,40 years taken in the chronological order: classes for 1979-1979 (1-year period), classes for 1979-1980 (2-year period), ... classes for 1979-2018 (40-year period). This input data used in chronological order is called "reference data".

However, the classification method may produce a different number of classes for data of the same volume but different years. Therefore, in order to produce estimations of class numbers that are robust to the choice of the data, we run additionally 60 classifications for the same data volumes of 1,2,…,40 years but picking the data randomly:

1) 30 classifications are built with data sampled randomly out of the whole data set (bootstrap method for data selection i.e. data elements may be repeated), cluster centres are initialized as described above in the method:

clusters with higher similarity are merged prior to those with lower similarity

2) 30 other classifications are built on the data selected randomly (but without repetitions) and cluster centres are initialized randomly: cluster pairs are merged randomly without the preference for more similar pairs (also in a case when the input data is the same as the "reference data", the random initialization of cluster centres yields different pathways of class merging)

The first group of the 30 classifications serves to prove the robustness of the classification method to the selection of the input data. The second group of the 30 classifications serves to illustrate the robustness to the initialization of clusters by the input data. We call both of two groups together "randomized data".

We expect that after a certain critical data amount is accumulated, further increase does not lead to a discovery of new classes and the temporal stability of the method is achieved. The minimum critical data amount, *minNYR* (=minimum number of years of data), is set when the number of resulting classes "levels out" and stabilizes.


The total 61 classifications (obtained on 1 "reference data" + 60 "randomized data") are compared to each other in the following way:

1) search for each class *i* of the classification *k* its counterpart (most similar class) *j* in the classification *l*: each pair of counterparts *(i,j)* is detected by maximizing *SSIM(i,j)* for all *i* and *j*;

2) weight the similarity value *SSIM(i, j)* by the frequency of *i* in the classification *k*: *SSIM(i, j)\*HIST(i)*, where *HIST(i)* is the relative frequency of class *i* in the classification *k*;

3) compute the total mean weighted similarity, *mwSSIM*, of the classification *k* to the classification *l* as the sum of weighted similarity values for all pairs of classes and their counterparts:

$$mwSSIM(k,l) = \sum_{i}^{N} SSIM(i,j) * HIST(i) \qquad (13)$$

where *N* – is the number of classes in the classification *k*, *i*=1,..*N*,

*j* – is the counterpart of class *i* (class *i* belongs to the classification *k*, class *j* is belongs to the classification *l* and is the most similar element to *i*),

*HIST(i)* – frequency of the class *i* in the classification *k*.

We compute the matrix of *mwSSIM* values using the 61 classifications retrieved on at least *minNYR* years of data (note: the

number *minNYR* is defined on the "reference data" as the minimum number of years of input data necessary to represent



possibly all classes i.e. further increase of this number does not increase the number of resulting classes). We require this matrix to have all elements $mwSSIM_{i,j}>0.40$, i.e. each pair of classifications derived on the same volume of data must be on average similar to each other. This "mean similarity" of the classifications indicates the temporal stability of the classes.

Cluster spatial stability. The stability of the method in space cannot be addressed by applying the clustering algorithm straightforwardly to the data on lower/higher spatial resolution because the pre-set threshold for cluster merging $TH_{merge}$ is not directly transferable to other spatial grids. The reason for this is simple: a pair of images at a high resolution that appears dissimilar to an observer may have similar low-resolution prototypes (when similarity-determining details are averaged out). However, it can be required that the method determines structurally similar classes at any spatial resolution. To test this, the clustering algorithm is run on the same data but of reduced (4º×6º) and increased (1°×1.5°) spatial resolution. The corresponding data sets were built by a resampling of the original data on the low-resolution (4°×6°) and on the high-resolution (1°×1.5°). The retrieved classes from these data sets are compared to the classes on the reference grid (2°×3º).

Cluster reproduction and representativity. The method uses medoids as cluster centres and, therefore, the resulting class representatives (set of medoids) are elements of the original data and are physically interpretable/plausible synoptic patterns. However, it is necessary to demand that a cluster medoid represents all cluster elements and their whole entity as a group. For each cluster, we compare the cluster centre (medoid) to the cluster mean (centroid) and calculate their similarity value. Based on the similarity values we analyse the representativity of the cluster elements by the medoids. We require that all medoids are strongly similar (SSIM>0.60) to their centroids. Representing a cluster by a medoid guarantees that the medoid has a minimum similarity to each of the cluster elements, furthermore, it is the element with the largest total similarity to all of cluster elements. If a centroid and a medoid of some class are dissimilar, this indicates that there is a group of elements in the class that are dissimilar to the medoid.

### 3.7 Statistics for model evaluation

The classification done on the "reference data" (reanalysis ERA-Interim of 1979-2018) yields the set of "reference SP-classes". Each data element of the reference data itself, of an alternative reanalysis data (NCEP1) and of each CMIP6-model are assigned to one of the "reference SP-classes" to which it has the maximal similarity. We suggest to compare different datasets assigned to the "reference SP-classes" using the following statistics: histogram of frequencies (*HIST*) for SP-classes through all years and seasons, histograms of frequencies for each season ($HIST_{DJF}$, $HIST_{MAM}$, $HIST_{JJA}$, $HIST_{SON}$), the matrix of transitions (*TRANSIT*) between available classes (frequency for each SP-class to follow another SP-class), and probability of persistence (*PERSIST*) of each SP-class for 1, 2,.. 25 days. Whereas, statistics *HIST*, $HIST_{DJF}$, $HIST_{MAM}$, $HIST_{JJA}$, and $HIST_{SON}$ are one-dimensional vectors with the number of components equal to the number of SP-classes, the *TRANSIT* and *PERSIST* are two-dimensional matrices. In case of high dimensionality i.e. many SP-classes, the comparison of these vectors and matrices may become awkward and ambiguous. Therefore for quantifying differences between pairs of such statistics we suggest to weight contributions of each class by its frequency. We compute Jensen-Shannon divergence (Eq.14, similar to the widely used Kullback–Leibler divergence but symmetric and it always has a finite value): frequent elements govern contributions to the





distance measure, and vice versa, rare elements make smaller contributions. The Jensen–Shannon divergence, *JSD*, used here
to measure the similarity between two probability distributions $P$ and $Q$ defined on the same probability space $\boldsymbol{\chi}$, is computed
in this way:

$$JSD(P \parallel Q) = \frac{1}{2}\sum_{x \in X} P(x) \ln \frac{P(x)}{M(x)} + \frac{1}{2}\sum_{x \in X} Q(x) \ln \frac{Q(x)}{M(x)} \qquad (14)$$

where the probability distributions $P$ and $Q$ are the normalized (to the sum of 1.0) frequency histograms, transition- and
persistence-matrices of the reference ($Q$) and a model ($P$); space $\boldsymbol{\chi}$ is a one- or two-dimensional space; $M$ is the mean
probability distribution:

$$M = \frac{P+Q}{2} \qquad (15)$$

It is common to compute the square root of *JSD* as a true metric for distance, Jensen–Shannon distance (Eq.16):

$$J(P \parallel Q) = \sqrt{JSD(P \parallel Q)} \qquad (16)$$

Such distance measure is robust against the "noise" from rare classes and as well as rare class-to-class transitions.

**3.8 Quality index**

We compute Quality Index (*QI*) that quantifies how well the synoptic circulation patterns are represented in the climate
simulation data as compared to the reference data. For each of the seven statistics an individual *QI* is computed as suggested
by Sanderson et al. (2015) but using the Jensen-Shannon distance as follows:

$$QI(P \parallel Q) = exp^{-a \sqrt{J(P \parallel Q)}} \qquad (17)$$

where the normalizing constant $a$=1 for each statistic.

The overall Mean Quality Index is then computed as the mean of the seven individual quality indices: *QI(HIST), QI(HIST$_{DJF}$),*
*QI(HIST$_{MAM}$), QI(HIST$_{JJA}$), QI(HIST$_{SON}$), QI(TRANSIT)* and *QI(PERSIST)*.

Benchmark: the overall Mean Quality Index computed for the alternative reanalysis data NCEP1 is expected to be higher than
for any CMIP6 model. We see this value as the estimate of the maximum achievable *QI* for the best performing model.

**4 Results**

**4.1. Synoptic classes, effect of the threshold *TH$_{merge}$* on the number of classes**

We run the classification algorithm on the "reference data" of consistently increasing data volume of 1, 2,.. 40 years and
perform 60 additional runs with the "randomized data" for the same data volumes. We repeat every run three times varying
the threshold *TH$_{merge}$* – the threshold on similarity between two SPs that defines when these SPs are merged into one class. In
total (1+ 60)*3 = 183 runs of classification algorithm, each yielding a set of classes, are available for the analysis. Figure 7
shows the evolution of the number of classes in dependence on the volume of input data for three values of *TH$_{merge}$*. Figure 7
illustrates the influence of tightening the requirement on similarity for building clusters: higher thresholds *TH$_{merge}$* produce




larger numbers of final classes with higher within-class similarity of its members. However, at the same time the higher $TH_{merge}$

also loosens the requirement to separation among classes (higher similarity between classes is possible).

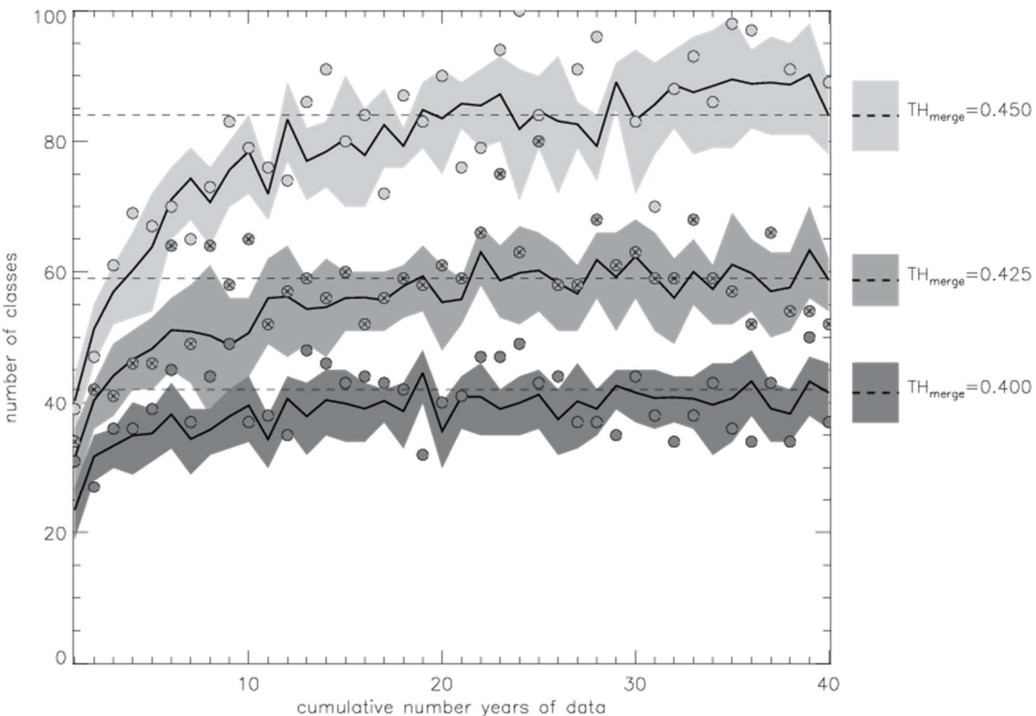

**Figure 7: Number of classes depends on the threshold $TH_{merge}$ and on the amount of clustered data. For each tested value of $TH_{merge}$**
**the black solid line shows the mean number of classes computed on 61 classifications (1 with "reference data" + 60 with "randomized data"); the shaded area shows the range of one standard deviation from the mean. The circles show numbers of classes from classifications with the "reference data"; circles with crosses highlight class numbers with $TH_{merge}$=0.425. The horizontal dashed lines show the mean number of classes for each $TH_{merge}$-value computed on the "reference data" of 40 years.**

The application of three values for the threshold $TH_{merge}$= 0.40, 0.425, and 0.45 to the "reference data" of maximal volume of

40 years produce 37, 52 and 89 classes, respectively. Computed on all 61 classifications (1 with "reference data" + 60 with

"randomized data") for varying $TH_{merge}$ the numbers of classes (mean ± standard deviation) are estimated 42±6, 59±4, and

84±5, respectively (Figure 7). As expected, the higher values of $TH_{merge}$ provide larger numbers of classes, although not larger

standard deviations of these numbers from their means, as a result of tightening the requirement for within-class similarity.





One of the features of our new two-stage clustering algorithm is that it classifies all synoptic patterns including rare ones. This
is the reason for the high number of classes build by this algorithm. Figure 8 shows the 37 classes built on the 40 years of the
"reference data" with *THmerge=0.40*: the six most frequent classes SP1, SP3, SP4, SP6, SP15 and SP27 represent together
~42% of the input data, 10 most rare classes (SP11, SP20, SP21, SP23, SP26, SP30, SP31, SP34, SP35, SP37) represent
together less than 5% of the input data.

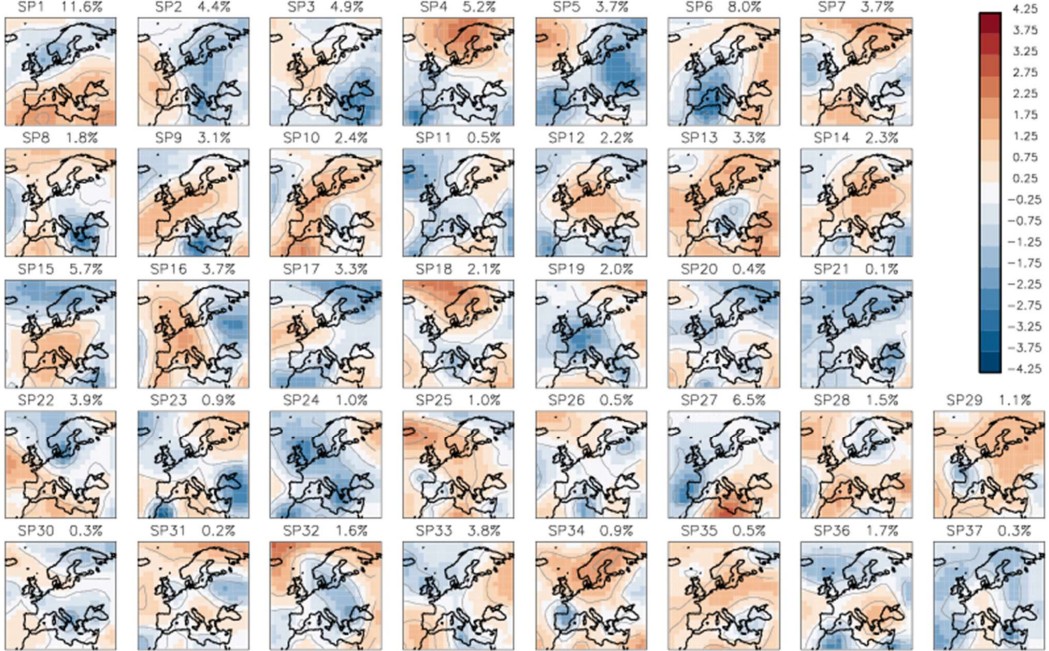

**Figure 8: SP-classes (anomalies of geopotential height) obtained on the "reference data" (ERA-Interim Reanalysis, 1979-2018) with**
**the threshold for similarity *THmerge=0.40*. Frequencies of SP-classes are shown above the corresponding plots.**

We take a closer look at the six most frequent SP-classes and their full fields (mean + anomaly) as shown in Figure 9. We
compare these six classes to the 29 synoptic weather patterns GWL-REA v1.3 fields (personal communication) developed in
German Meteorological Service (Deutscher Wetterdienst, www.dwd.de ) - Hess-Brezowksy Grosswetterlagen identified on
reanalysis data based on correlations in combination with Lamb Weather Type statistics (James and Ostermöller, 2022). For
each of the six SP-classes we compare its similarity value to each of the GWL-REA v1.3 field (Geopotential) and identify the
most similar one/pair:

-    SP1:  Cyclonic South-Westerly (SWZ- Südwestlage zyklonal)/Cyclonic Westerly (WZ - Westlage zyklonal)
-    SP6: Low over Central Europe (TM- Tief Mitteleuropa)
620         -    SP27: Low over the British Isles  (TB - Tief Britische Inseln)



- SP15: Anticyclonic Westerly (WA- Westlage antizyklonal)
- SP4: Anticyclonic South-Easterly (SEA - Südostlage antizyklonal)
- SP3: Anticyclonic North-Westerly (NWA - Nordwestlage antizyklonal)

Correspondences of the six frequent classes to the patterns GWL-REA v1.3 provide us with an evidence that, albeit not tuned to and not required to mimic semi-manual classifications, the new classification method determines not just arbitrary synoptic patterns but meaningful synoptic situation described by experts.

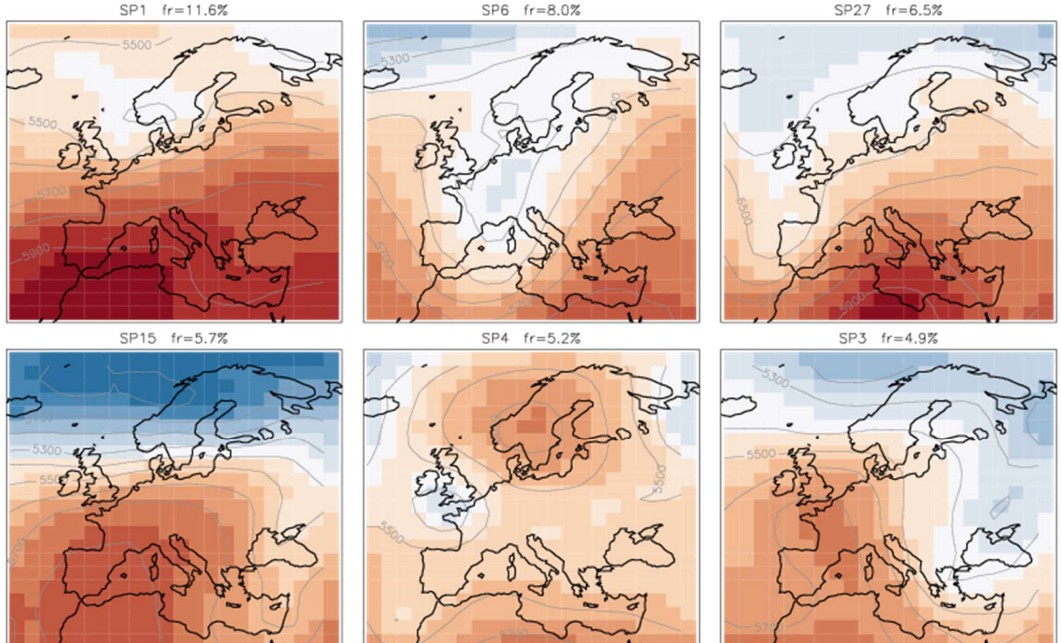

**Figure 9: Geopotential Height [m] for six most frequent SP-classes. The contour lines show the geopotential height levels every 100 m (labelled). The number of the SP-class, its frequency and the corresponding calendar date are given above each plot.**

The three sets of classes obtained on the "reference data" of the full volume with varying $TH_{merge}$ are further analysed with respect to consistency, separability, stability, and representativity of the data.

**4.2 Cluster consistency**

The evolution of classes built with different values of $TH_{merge}$ is presented in the form of a diagram (Figure 10), which is also called "arrow diagram" suggesting that lines show how classes are related among different sets of classes. For the "arrow



diagram" in Figure 10 the classes are derived by running the clustering algorithm on the data of one full year. We chose this minimal data volume to build classes with few elements for demonstrating the tightening similarity constrain (by the threshold $TH_{merge}$) in the best way as classes with large numbers of elements may reveal similarities among subsets of some elements

and overload the diagram. In Figure 10 identical classes (SSIM=1 for the medoids) are connected with thick solid black lines, strongly similar classes ($0.60 \leq SSIM < 1$) are connected with dashed thick black lines, similar classes ($0.40 \leq SSIM < 0.60$) are connected by thin grey lines, where connections with $0.40 \leq SSIM < 0.425$ are dashed. When increasing the merging threshold $0.40 \rightarrow 0.425$ the total number of classes rises $31 \rightarrow 34$ with 26 classes remaining identical or "strongly similar", 5 remain without a strongly similar counterpart and 8 new classes emerge. Further rising the threshold value $0.425 \rightarrow 0.45$ leads to

building of 39 classes with 36 classes remaining identical or strongly similar, 2 classes remain without a strongly similar counterpart and 7 new classes emerge. The new emerging classes may have similarity to more than one previous class. We see that 23 classes retain their medoids through the two steps of tightening the similarity constrain ($0.40 \rightarrow 0.425 \rightarrow 0.45$). It is important to note: the identical classes have only one counterpart in each set of classes that means they are "transferred" to the next set of classes obtained with a higher $TH_{merge}$ and not "split" into new classes. The "strongly similar" classes typically have

only one or - rarely - only few counterparts i.e. they are rarely split. New emerging classes may have similarities to multiple original classes. The fulfilment of the demand on the consistency of class evolution is shown by the prevalence of identical classes in the diagram, indicating one-to-one correspondence between classes of different sets. The identical classes, which remain unchanged, are connected with thick solid lines and are often accompanied by a 'bunch' of thin lines. Such 'bunches' are mainly produced by the breaking off some elements from the class on the left side into its own class on the right side; the

medoid of the original class on the left side remains preserved. For new emerging classes (on the right side) similarities to multiple original classes (on the left side) are acceptable as new classes may contain elements broke off from multiple original left side classes. An unwanted form of the diagram would be a distribution of classes from set to set connected with thin lines, without clearly preserved identical types.





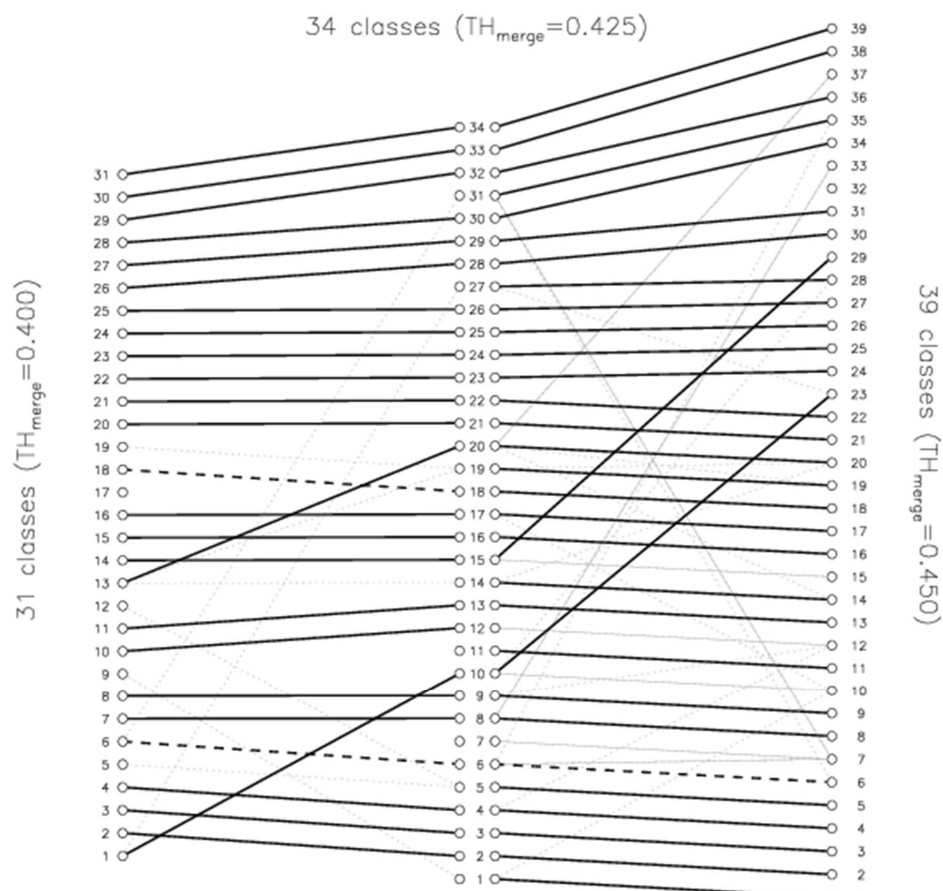

**Figure 10: Similarity between classes derived with different merging threshold: (left) 31 classes obtained with $TH_{merge}=0.40$, (middle) 34 classes with $TH_{merge}=0.425$, and (right) 39 classes with $TH_{merge}=0.45$. Black thick lines connect identical classes ($SSIM=1$), black dashed lines connect "strongly similar" classes ($0.60≤SSIM<1$), grey lines connect similar classes ($0.40≤SSIM<0.60$), where connections with $0.40≤SSIM<0.425$ are dashed. Following the black solid lines from left-to-right: 23 classes retain their medoids.**

### 4.3 Cluster separability

The metrics *EV*, *DRATIO* and indicators $SSIM_{in}$, $SSIM_{out}$, *SSIMRATIO* computed on the classes obtained with increasing $TH_{merge}$ illustrate the importance of the choice of this threshold and its influence on the number of derived classes and their separability. Table 1 presents the values of the chosen metrics and indicators. Please note: metrics *EV* and *DRATIO* illustrate



only (!) the influence of the $TH_{merge}$ on the final set of classes and do not describe the quality of classes as they are computed using the Euclidean Distance – a measure that was not optimized by the clustering algorithm. Therefore, $EV$ and $DRATIO$

should not be used to assess the absolute performance of the classification, but the relative performance depending on $TH_{merge}$. Classifications with larger numbers of classes achieve a better skill $EV$ than those with less classes due to the natural fact that a larger number of classes captures a higher fraction of the variation. The extreme case, when the total variation is explained completely ($EV=1$), is achieved when the number of classes is equal to the number of data. Therefore, it would be dangerous to favour classifications with larger numbers of classes based on this metric. In the present study, the set of classes obtained

with $TH_{merge}=0.45$ provides the highest ratio of explained variation. Clusters of randomly chosen groups, as expected, show nearly no explained variation at all (see Table 1).

Values of the metric $DRATIO$ <1.0 indicate that, on average, elements within classes have shorter Euclidean distance to each other than to elements of other classes. Smaller values of $DRATIO$ indicate a stronger separation of classes. The highest value of $TH_{merge}=0.45$ provides the lowest value of $DRATIO$ and, therefore, shows the best separation of classes in terms of Euclidean

distance. In randomly chosen groups the value of $DRATIO$ is close to 1, as also shown in Table 1, because of nearly equal distances between elements of the same class and of different classes.

**Table 1: Metrics for classes obtained in three experiments with varying merging-threshold ($TH_{merge}$) applied on the "reference data" of 40 years. Values after "/" are those computed on random groups.**

| $TH_{merge}$ | Number of classes | $EV$ classes/random | $DRATIO$ classes/random | $SSIM_{in}$ classes/random | $SSIM_{out}$ classes/random | $SSIMRATIO$ classes/random |
|---|---|---|---|---|---|---|
| 0.40 | 37 | 0.3825/0.0028 | 0.6059/0.9968 | 0.3252/0.0317 | 0.0158/0.0316 | 20.58/1.00 |
| 0.425 | 52 | 0.4055/0.0042 | 0.5839/0.9954 | 0.3412/0.0319 | 0.0180/0.0316 | 19.96/1.01 |
| 0.45 | 89 | 0.4476/0.0066 | 0.5447/0.9929 | 0.3695/0.0317 | 0.0215/0.0316 | 17.19/1.00 |

Indicators $SSIM_{in}$ and $SSIM_{out}$ represent the influence of the similarity constrain by $TH_{merge}$ on the separability/homogeneity of the final classes. A good performance of the classification is achieved when similarity among elements of one class $SSIM_{in}$ is much higher than the similarity to elements of other classes $SSIM_{out}$ i. e. $SSIMRATIO$ should be maximized. The maximal mean similarity among elements of the same class ($SSI_{in}=0.3695$) is given by $TH_{merge}=0.45$, however, the mean similarity between pairs of elements of different classes ($SSI_{out}=0.0215$) is also the highest for this threshold indicating stronger

similarities among elements of different classes as well. Finally, $SSIMRATIO$ – an indicator of class separation on terms of similarity - is highest (20.58) for $TH_{merge}=0.40$ and shows the favourable separation of classes in terms of similarity among elements.

**At this point we make an important decision and choose the classification obtained with the merging threshold of $TH_{merge}=0.400$ for further analysis for two reasons: 1) this threshold provides good class separation; 2) using this value**



**we produce fewer classes, which can be meaningfully statistically analysed (a higher threshold value would produce more classes with fewer members). It is also important to note that a smaller number of classes is easier to describe verbally, more intuitive to understand and to separate visually.**

The stop-criterion in the clustering algorithm guarantees that the maximum similarity between final classes is less than $TH_{merge}$. In other words, there is no pair of final medoids similar to each other, otherwise they would have ended up in the same cluster.

Although it cannot be demanded that cluster centroids (means) also satisfy the same criterion on the maximum pairwise similarity, it can be demanded that cluster centroids are at least not "strongly similar" i.e. pairwise $SSIM<0.60$. Figure 11 shows matrices of pairwise similarities for medoids (left) and for corresponding centroids (right). Some pairs of centroids have a similarity value higher than any pair of medoids (circles show $SSIM \geq 0.40$) due to the fact, that the similarity of medoids but not of centroids was the optimized quantity in the clustering algorithm. The maximal similarity for a pair of centroids is

$SSIM=0.542$ (for centroids 1 and 22) i.e. there is no pair of "strongly similar" centroids. This gives an evidence that the two-stage clustering algorithm that uses medoids as class centres produces classes with also meaningfully separated centroids.

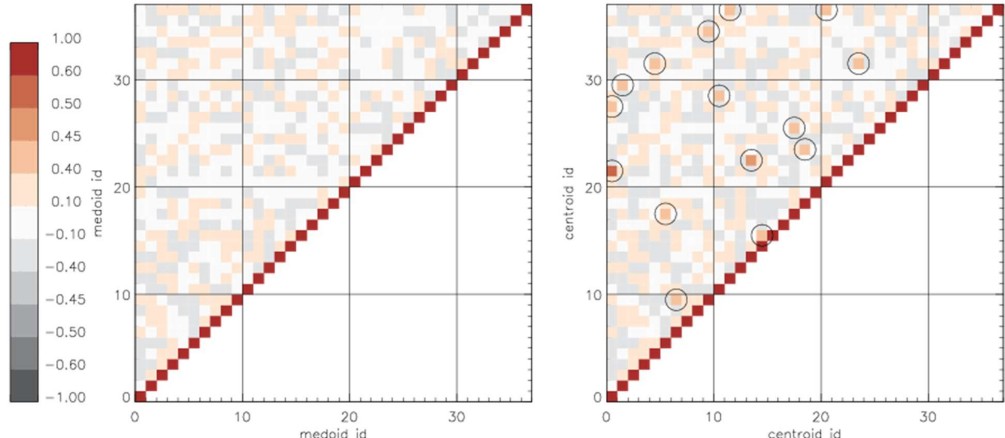

**Figure 11: Matrix of pairwise similarity values for 37 classes derived with $TH_{merge}=0.40$. Left panel: the matrix of SSIM for cluster**
**medoids. Right panel: the matrix of SSIM for cluster centroids. Circles show similarity values greater than 0.40. The only upper left half of each matrix is shown due to the symmetry; diagonal elements have SSIM=1.**

### 4.4 Cluster stability

Temporal stability. As we apply the classification algorithm on the data volume of 1,2, .. 40 years. The number of derived classes "levels off" after approximately 30 years of daily data for all values of $TH_{merge}$ (Figure 7): this means that all possible

synoptic patterns are likely to be captured within 30 years. This data volume matches with periods typically used for assessing the variability of other climate variables. Thus, we recommend the minimum critical data amount $minNYR=30$ years of data





for a temporally stable classification. To support this recommendation, we compute the matrix of the "mean weighted similarity" $mwSSIM$ for 61 sets of classes retrieved on 30, 35 and 40 years of data. We require this matrix to have all elements $mvSSIM_{i,j} > 0.40$ i.e. each pair of sets of classes must be on average similar to each other.

The number of classes in all 61 sets generated on $minNYR=30$ years of data varies from 36 to 59 classes, with the mean number of classes 42. For all 61 sets of classes, we computed the pairwise mean weighted similarity $mwSSIM$ (Figure 12). The value of $mwSSIM(k,l)$ shows the match of all classes from the set $k$ to all classes from the set $l$, weighted by the frequency of the classes in the set $k$. The matrix of pairwise $mwSSIM$ values is not symmetric: $mwSSIM(k,l) \neq mwSSIM(k,l)$ as the sets $k$ and $l$ may have different numbers of classes and also the classes differ. When the numbers of classes in sets $k$ and $l$ are different, the

following may occur: for class $i$ from set $k$ the class $j$ from set $l$ is the most similar counterpart, but (!) for the class $j$ from the set $l$ a different class $h$ from set $k$ is the most similar one, leaving the class $i$ being the second most similar counterpart for $j$. In a case of a "perfect match" the $mwSSIM=1$ i.e. indicating the identity of two sets of classes. Negative values of $mwSSIM$ would indicate two different sets of classes without any element from one set similar to any element in the other set. In our analysis we only consider $mwSSIM$ for different pairs of classifications (diagonal elements of the $mwSSIM$ matrix are always 1.0

anyway). The maximum $mwSSIM = \sim 1.00$ (almost identity!) is attained by 7% of all pairs, strong similarity with $mwSSIM \geq 0.60$ show 54% of all pairs. The mean value of pairwise $mwSSIM$ for all [different] classifications is $0.63$. Figure 12 shows that all sets of classes are similar to all sets of classes in a pairwise comparison i.e. all pairwise similarity values are greater than the threshold ($TH_{merge}=0.40$) with the minimum $mwSSIM =0.53$. This is indeed a good result. This allows us to say that the two-stage classification produces similar sets of clusters also when initialized by randomly chosen subsets of the input

data.





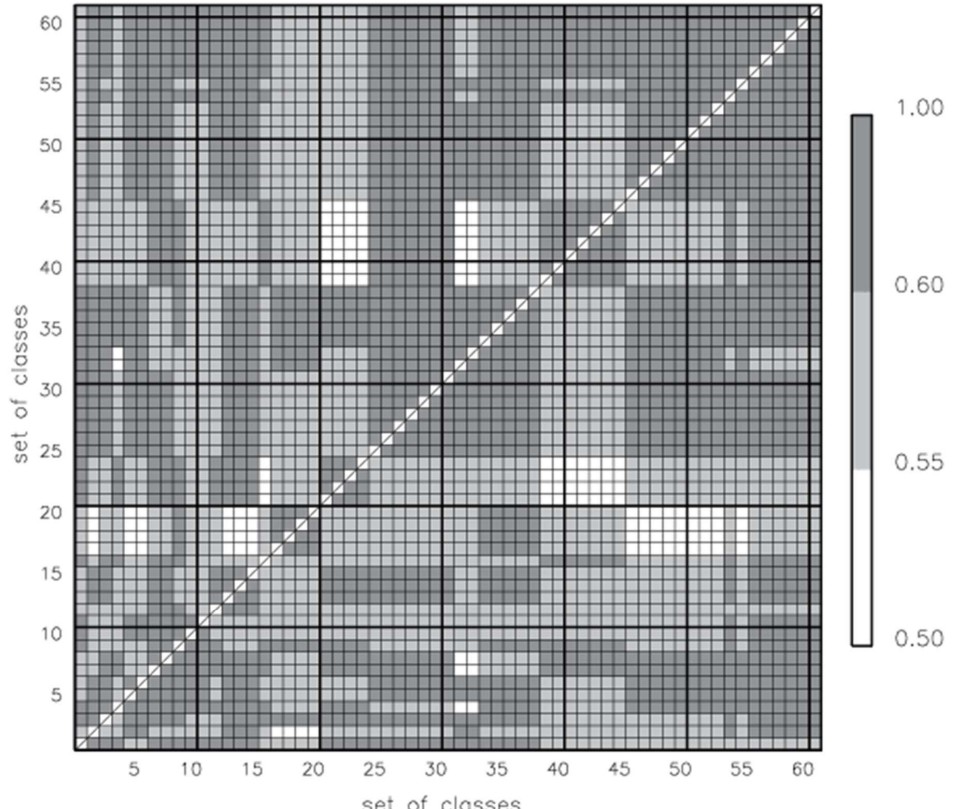

**Figure 12: Mean weighted similarity, *mwSSIM*, for each of 61 sets of classes to each other. Each set of classes was derived on *minNYR* =30 years of input data. The diagonal elements are not shown because *mwSSIM=1* of a set of the classes to itself. The matrix is not symmetric: *mwSSIM(k,l)≠mwSSIM(k,l)* as the sets of classes *k* and *l* may have different number of classes.**

We repeat the calculation of *mwSSIM* on 35 and 40 years of data (not shown) in order to make sure that the classification algorithm produces similar sets of classes on larger data volumes as well. When the data volume is set *minNYR*=35 the number of classes varies from 31 to 48 among 61 sets of classes, with minimum *mwSSIM = 0.55* and mean *mwSSIM = 0.65*. On the maximal data volume (40 years) number of classes varies from 35 to 49, with minimum *mwSSIM = 0.54* and mean *mwSSIM = 0.64*. These calculations of *mwSSIM* on other data volumes only support our previous findings: all pairwise values of



*mwSSIM* are greater than the similarity threshold indicating that our two-stage clustering algorithm applied to randomly chosen data builds sets of similar classes.

Spatial stability. For testing the stability of the method in space, additionally to the classes on the reference data set (2ºx3º), two sets of classes were built on the low-resolution (4ºx6º) and on the high-resolution (1ºx1.5º) by resampling the original reanalysis fields to these spatial resolutions. The clustering algorithm was run with the data on each spatial resolution using the same threshold *TH_merge=0.40*. This poses some restrictions on the interpretation of the results. First: two images on different spatial resolutions derived from the same original image are not necessarily identical (!) in terms of SSIM because they contain different amounts of information. The SSIM-value deteriorates with the increasing spatial resolution as the degree of detail in the images grows. Following this argument, it would be impossible to build the same set of classes at various spatial resolutions with the same threshold on similarity. However, it can be required that some classes emerge at all spatial resolutions. Examples of such SP-classes are shown in Figure 13 at three spatial resolutions.





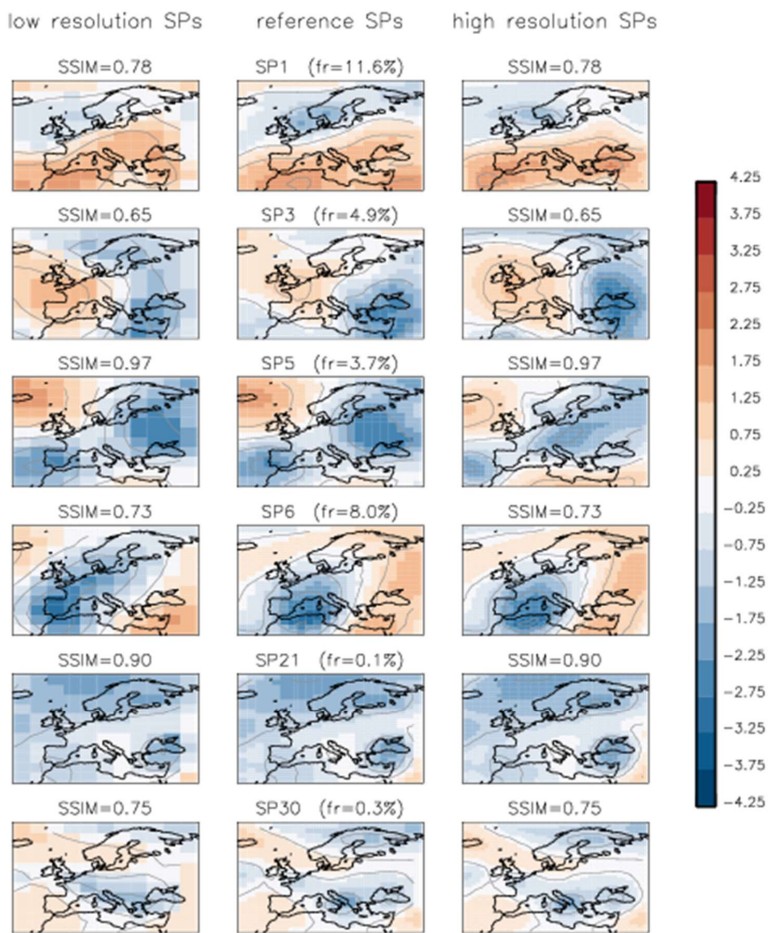


**Figure 13: Examples of SP-classes on three spatial resolutions. The middle column shows reference SPs built from the "reference data" (2°x3°, 22x22 grid cells) with their frequencies (%). The left-side column shows corresponding patterns on the low-resolution (4°x6°, 11x11 grid cells), the right-side column - on the high-resolution (1°x1.5°, 44x44 grid cells). Both plots for low- and high-resolution counterparts show the SSIM-value to the reference SP-class on top of each plot.**




Figure 13 shows six SP-classes at the original resolution (middle plots) and their counterparts in the low- and high-resolution sets of classes. Please note: the SP-classes are built at each resolution independently and are not just re-sampled copies of the same classes. Therefore, some discrepancy must be tolerated among the classes at different resolutions as they are medoids of
independently formed classes. Despite of such discrepancies the SP-classes show essentially the same geopotential anomalies at all spatial resolutions. Although it is not required, classifications on the three spatial resolutions have 37 classes each. The mean similarity for all 37 SP-classes built on the "reference data" to their counterparts on the low-resolution is 0.53; to their counterparts on the high-resolution is 0.52. These high numbers (>0.40) indicate the ability of the new clustering algorithm to reproduce similar SP-classes on different spatial resolutions i.e. spatial stability of this algorithm.

**4.5 Cluster reproduction and representativity**

The two-stage classification method uses medoids for representing clusters for the reasons of stability. A medoid of a cluster can be seen as "the representative element" of this cluster i.e. element most similar to all other elements in the cluster (definition of the medoid). Once the cluster is changed (merged with another one by the hierarchical step for example) the medoids are recomputed. Every new attribution of an element to a cluster is done to the most similar medoid (this ensures exclusive
attribution of similar elements to clusters). For the final set of classes we demonstrate that the medoids are strongly similar to cluster means (centroids) i.e. cluster medoids effectively represent the mean patterns of their classes. We analyse the set of 37 classes built on the "reference data" and compute for each class the similarity value between its centroid and medoid (Table 2). A good representativity is achieved when medoid and centroid of each class are "strongly similar". The Table 2 shows exactly this: $SSIM(medoid_i, centroid_i) \geq 0.60$ for all classes $i$. If the medoid and the centroid are "strongly similar", it guarantees
that there are no or negligibly few "extravagantly" dissimilar members in that class. Otherwise, the mean (centroid) would have lost its similarity to the medoid being distorted by the averaging with dissimilar members. The "strong similarity" between medoids and centroids for all 37 classes was found indicating the very good representability of clusters by their medoids. The mean similarity between medoids and centroids over all 37 classes is 0.78; weighted by the class frequency is 0.79. This is a very good result that shows the strong resemblance between medoids and centroids of the clusters and illustrates the
representativity of classes by their medoids.

**Table 2: Set of 37 SP-classes on "reference data" with $TH_{merge}=0.40$: index of Synoptic Pattern (SP), Fraction (Fr) in percent of the class in the "reference data" and the Similarity (SSIM) value between medoid and centroid of the class.**

| SP | Fr [%] | SSIM | SP | Fr [%] | SSIM | SP | Fr [%] | SSIM | SP | Fr [%] | SSIM |
|----|--------|------|----|--------|------|----|--------|------|----|--------|------|
| 1 | 11.6 | 0.77 | 11 | 0.5 | 0.71 | 21 | 0.1 | 0.67 | 31 | 0.2 | 0.75 |
| 2 | 4.4 | 0.85 | 12 | 2.2 | 0.79 | 22 | 3.9 | 0.77 | 32 | 1.6 | 0.76 |
| 3 | 4.9 | 0.81 | 13 | 3.3 | 0.78 | 23 | 0.9 | 0.70 | 33 | 3.8 | 0.84 |
| 4 | 5.2 | 0.81 | 14 | 2.3 | 0.83 | 24 | 1.0 | 0.76 | 34 | 0.9 | 0.73 |
| 5 | 3.7 | 0.80 | 15 | 5.7 | 0.80 | 25 | 1.0 | 0.76 | 35 | 0.5 | 0.81 |





| 6 | 8.0 | 0.79 | 16 | 3.7 | 0.76 | 26 | 0.5 | 0.84 | 36 | 1.7 | 0.76 |
|---|-----|------|----|-----|------|----|-----|------|----|-----|------|
| 7 | 3.7 | 0.81 | 17 | 3.3 | 0.80 | 27 | 6.5 | 0.76 | 37 | 0.3 | 0.74 |
| 8 | 1.8 | 0.79 | 18 | 2.1 | 0.82 | 28 | 1.5 | 0.72 | - | - | - |
| 9 | 3.1 | 0.77 | 19 | 2.0 | 0.74 | 29 | 1.1 | 0.75 | - | - | - |
| 10 | 2.4 | 0.77 | 20 | 0.4 | 0.73 | 30 | 0.3 | 0.79 | - | - | - |

Figure 14 illustrates medoids and centroids for the five most frequent SP-classes. As expected, each medoid has a higher
amplitude of anomalies and the corresponding centroid shows essentially the same pattern but with weaker anomalies. The
Mean Absolute Difference (the sum of absolute differences between each element in the class and its medoid) between the
two shows the highest values at the locations of strong amplitudes in the medoid fields and lower values at locations on "edges"
of synoptic patterns. This is expectable because the covariance term of SSIM (Eq. 3) penalizes a displacement of anomalies
stronger than a mismatch of anomalies' amplitudes i.e. steers the clustering method to prefer correctly placed anomalies over
their correctly estimated amplitudes (by a false placement). This illustrates that the clustering method sensitively groups SP-
patterns with similar composition of the anomalies into classes using the SSIM as a similarity measure for pairs of geopotential
fields.





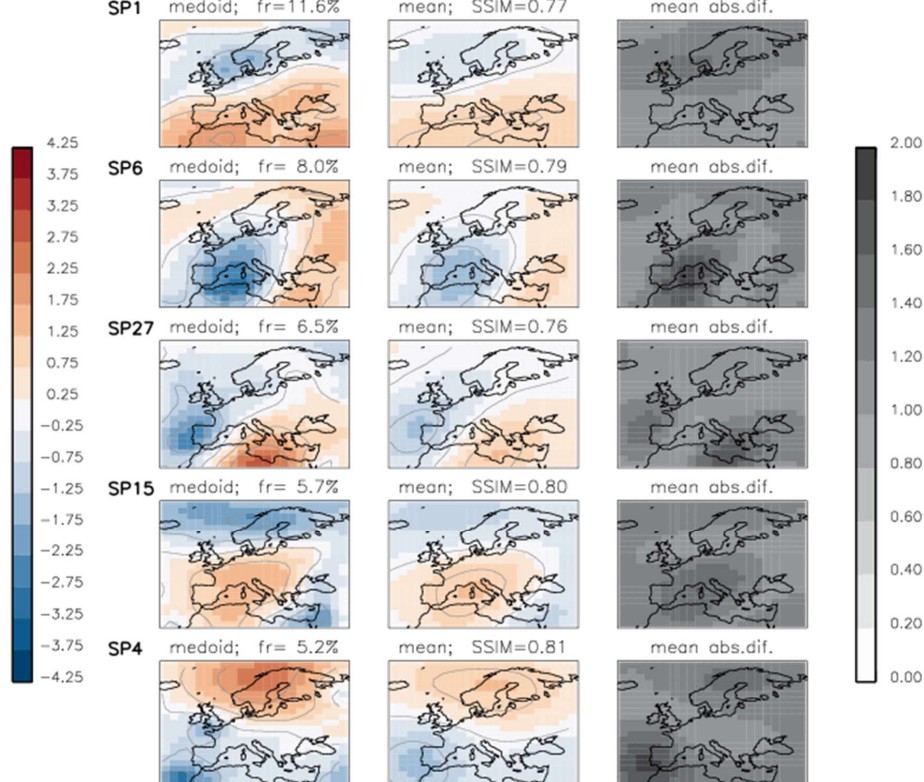

**Figure 14: Medoids (left column), means/centroids (centre column) and their Mean Absolute Difference (right column) for five most frequent SP-classes. Frequency of each SP-class is shown on top of each medoid plot, SSIM between medoid and centroid is shown on the top of each plot of the mean.**

## 5 Statistics and Quality indices for evaluation of CMIP6 historical climate simulations

In Figure 15 we show examples of the three statistics: histogram of class frequencies HIST, class-to-class transition matrix TRANSIT, and matrix of persistence PERSIST of each class for 1,2,..8 days. We chose to present these statistics for only three data sets - the reference and two models - for demonstration purposes. Figure 15a shows the large spread of frequencies of SP-classes that conditions high spread in frequencies of transition matrices (Figure 15b) and persistence matrices (Figure 15c). As we aim to evaluate CMIP6 climate historical simulations, we assign each models output to the set of reference SP-classes and compute the distance metric for this model to the reference using the Jensen–Shannon distance (Eq.12).



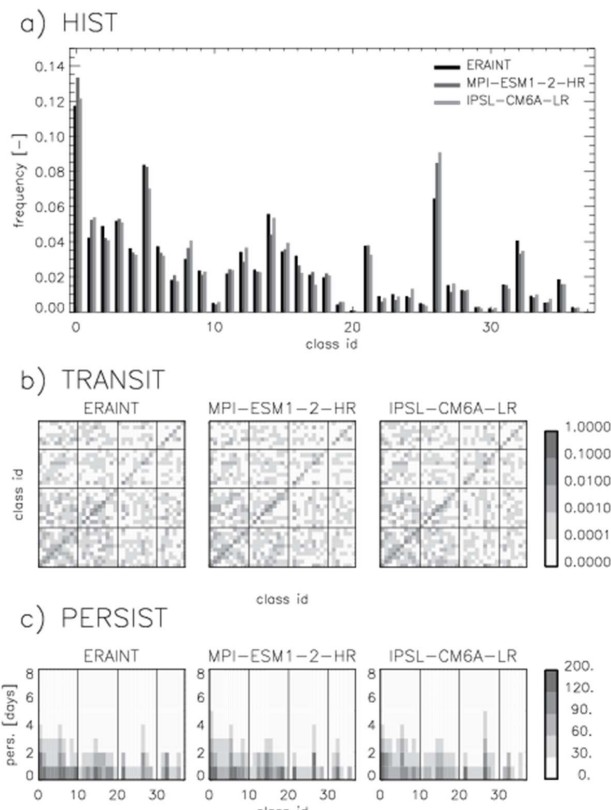

**Figure 15: Examples of statistics for the reference data and two CMIP6-models: a) HIST – histogram of frequencies for each SP-class; b) TRANSIT matrices for each pair of SP-classes; where diagonal elements show transition for the same class i.e. persistence, c) PERSISTENCE matrices that show number of events when each SP-class persisted for 1,2, ..8 days (this statistic is shown in absolute values - non-normalized – for better readability).**

The Jensen–Shannon distance is computed for one-dimensional statistics ($HIST$, $HIST_{DJF}$, $HIST_{MAM}$, $HIST_{JJA}$, $HIST_{SON}$) as well as for two-dimensional $TRANSIT$, $PERSIST$ between the two probability distributions is computed for each model and the reference.

Finally, we compute seven Quality Indices (Table 3) that can be combined to suit objectives of the model evaluation, for example, seasonally separated $QI(HIST_{DJF})$, $QI(HIST_{MAM})$, $QI(HIST_{JJA})$, $QI(HIST_{SON})$ can be used in evaluating seasonal frequencies of synoptic patterns, $QI(PERSIST)$- for evaluating of the duration of synoptic patterns. In this paper, we equally weight all $QI$s and compute the Mean Quality Index (Table 3). A quality index of 1 indicates the identity between the model





and the reference. The benchmark for this study: the Mean Quality Index for an alternative reanalysis NCEP1 is 0.84 and can
be viewed as the best possible *QI* for a model.

**Table 3: CMIP6 Models and their Quality Indices (*QI*). The mean Quality Index (*Mean QI*) is computed for each model as the mean of its individual *QIs* for each model statistic. The two last rows contain the mean (*MEAN*) and the standard deviation (*STDDEV*) of**
**all *QIs* for the same statistic across 32 CMIP6 models.**

| Nr | Model name | QI for individual statistics | | | | | | | Mean QI |
|---|---|---|---|---|---|---|---|---|---|
| | | HIST | HIST$_{DFJ}$ | HIST$_{MAM}$ | HIS$_{JJA}$ | HIST$_{SON}$ | TRANSIT | PERSIST | (all QIs) |
| - | ERAINT(ref.reanalysis) | 1,00 | 1,00 | 1,00 | 1,00 | 1,00 | 1,00 | 1,00 | 1,00 |
| - | NCEP (alt.reanalysis) | 0,89 | 0,88 | 0,87 | 0,85 | 0,87 | 0,75 | 0,78 | 0,84 |
| 1 | ACCESS-CM2 | 0,79 | 0,71 | 0,77 | 0,70 | 0,75 | 0,67 | 0,70 | 0,73 |
| 2 | AWI-ESM-1-1-LR | 0,76 | 0,73 | 0,74 | 0,70 | 0,71 | 0,66 | 0,70 | 0,72 |
| 3 | BCC-CSM2-MR | 0,78 | 0,73 | 0,75 | 0,69 | 0,72 | 0,66 | 0,71 | 0,72 |
| 4 | BCC-ESM1 | 0,77 | 0,71 | 0,72 | 0,69 | 0,72 | 0,66 | 0,70 | 0,71 |
| 5 | CanESM5 | 0,78 | 0,70 | 0,73 | 0,74 | 0,73 | 0,66 | 0,70 | 0,72 |
| 6 | CESM2 | 0,78 | 0,74 | 0,75 | 0,71 | 0,73 | 0,67 | 0,70 | 0,72 |
| 7 | CESM2-FV2 | 0,76 | 0,70 | 0,74 | 0,69 | 0,71 | 0,65 | 0,69 | 0,71 |
| 8 | CESM2-WACCM-FV2 | 0,76 | 0,71 | 0,71 | 0,68 | 0,74 | 0,66 | 0,70 | 0,71 |
| 9 | CMCC-CM2-SR5 | 0,76 | 0,72 | 0,75 | 0,67 | 0,73 | 0,66 | 0,70 | 0,71 |
| 10 | CNRM-CM6-1 | 0,78 | 0,72 | 0,75 | 0,68 | 0,74 | 0,66 | 0,70 | 0,72 |
| 11 | CNRM-ESM2-1 | 0,81 | 0,73 | 0,74 | 0,71 | 0,74 | 0,67 | 0,70 | 0,73 |
| 12 | EC-Earth3 | 0,79 | 0,74 | 0,76 | 0,69 | 0,73 | 0,67 | 0,71 | 0,73 |
| 13 | EC-Earth3-Veg | 0,77 | 0,74 | 0,75 | 0,67 | 0,75 | 0,66 | 0,71 | 0,72 |
| 14 | FGOALS-f3-L | 0,77 | 0,68 | 0,72 | 0,66 | 0,76 | 0,66 | 0,70 | 0,71 |
| 15 | FGOALS-g3 | 0,76 | 0,69 | 0,73 | 0,68 | 0,75 | 0,66 | 0,69 | 0,71 |
| 16 | GISS-E2-1-G | 0,78 | 0,70 | 0,73 | 0,66 | 0,74 | 0,66 | 0,71 | 0,71 |
| 17 | HadGEM3-GC31-LL | 0,80 | 0,72 | 0,76 | 0,72 | 0,75 | 0,67 | 0,70 | 0,73 |
| 18 | HadGEM3-GC31-MM | 0,79 | 0,74 | 0,75 | 0,71 | 0,76 | 0,67 | 0,71 | 0,73 |
| 19 | INM-CM4-8 | 0,77 | 0,72 | 0,73 | 0,66 | 0,72 | 0,65 | 0,69 | 0,71 |
| 20 | INM-CM5-0 | 0,78 | 0,74 | 0,73 | 0,71 | 0,70 | 0,67 | 0,69 | 0,72 |
| 21 | IPSL-CM6A-LR | 0,77 | 0,73 | 0,73 | 0,65 | 0,70 | 0,66 | 0,70 | 0,71 |
| 22 | IPSL-CM6A-LR-INCA | 0,79 | 0,70 | 0,74 | 0,66 | 0,70 | 0,66 | 0,69 | 0,71 |
| 23 | KACE-1-0-G | 0,80 | 0,74 | 0,75 | 0,70 | 0,75 | 0,67 | 0,70 | 0,73 |
| 24 | MIROC6 | 0,78 | 0,72 | 0,76 | 0,69 | 0,74 | 0,67 | 0,69 | 0,72 |
| 25 | MPI-ESM-1-2-HAM | 0,78 | 0,72 | 0,75 | 0,70 | 0,72 | 0,66 | 0,71 | 0,72 |
| 26 | MPI-ESM1-2-HR | 0,79 | 0,72 | 0,75 | 0,73 | 0,74 | 0,67 | 0,71 | 0,73 |
| 27 | MPI-ESM1-2-LR | 0,79 | 0,73 | 0,77 | 0,72 | 0,75 | 0,67 | 0,70 | 0,73 |
| 28 | MRI-ESM2-0 | 0,80 | 0,74 | 0,73 | 0,71 | 0,75 | 0,67 | 0,71 | 0,73 |
| 29 | NorESM2-LM | 0,76 | 0,70 | 0,69 | 0,66 | 0,70 | 0,65 | 0,69 | 0,69 |
| 30 | NorESM2-MM | 0,77 | 0,72 | 0,74 | 0,70 | 0,70 | 0,66 | 0,70 | 0,71 |
| 31 | TaiESM1 | 0,78 | 0,71 | 0,74 | 0,71 | 0,74 | 0,67 | 0,69 | 0,72 |
| 32 | UKESM1-0-LL | 0,78 | 0,76 | 0,75 | 0,69 | 0,74 | 0,67 | 0,70 | 0,73 |
| - | *MEAN (32 models)* | *0,78* | *0,72* | *0,74* | *0,69* | *0,73* | *0,66* | *0,70* | *0,72* |
| - | *STDDEV(32 models)* | *0,01* | *0,01* | *0,01* | *0,02* | *0,02* | *0,01* | *0,01* | *0,01* |





The mean quality index indicates how well the respective model captures the synoptic circulation in the reference data ERA-Interim. This quality index together with quality indices for scalar variables can be used for ranking the climate model simulations and as an evaluation measure. For example, the climate simulation NorESM2-LM seems to underperform all other

models (*Mean QI*=0.69) whereas other models have higher values. Such diagnostic is a useful complement for model evaluation: poor quality scores from evaluation of synoptic patterns should be seen as warning prior to analysing scalar variables.

## 6 Conclusions

We presented a new two-stage classification method that uses the Structural Similarity Index Measure (SSIM) for building

classifications of synoptic circulation patterns, which are described by geopotential anomaly at the level of 500hPa. This classification method produces a set of well separated, consistent, and representative classes. The algorithm demonstrated its robustness against temporal variability and to the spatial resolution of the data. It classifies all input data fields without pre-filtering and pre-initialization of classes, it builds structurally different classes with inter-class homogeneity. While explaining the procedure of developing the two-stage classification algorithm, we demonstrated the disadvantage of using the classical

clustering algorithm *k-means* and the MSE as distance measure for cluster building when classifying meteorological fields such as geopotential. We hope this demonstration helps users and developers of classification methods to be careful with interpreting their results and to be conscious that some problems (such as "snowballing") may be avoided by simple modifications of the clustering algorithm as illustrated in this paper.

The important strength of the new classification method - its applicability to any region on the globe with no requirements on

prior knowledge about weather types at that region. The applicability of our classification method to any region allows evaluation of models quasi-globally as it is done by Cannon (2020) for evaluation of CMIP5 and CMIP6 models: in 6 continental-scale regions (or more).

In this paper we describe the method – the recipe - to build a set of synoptic classes. We do not propose an "optimal classification" of synoptic patterns for all purposes. We apply the new method on the reanalysis data ERA-Interim and built a

set of synoptic classes (application of the classification method on other data sets may build other sets of synoptic classes). As an example, we use this set of classes to evaluate the performance of 32 global CMIP6 climate models in the CORDEX-EU domain. Model data were attributed to the reference set of classes and statistical parameters (frequency of occurrence of each pattern, frequency of transitions from one pattern to another, persistence of each pattern) were computed for these models. We compared these statistical parameters to the parameters computed on the reference data, calculating the Quality Index, and

suggest using the Quality Index in evaluation routines for climate models as an additional diagnostic measure. Using Quality Indices proposed in this study would help to avoid misinterpretations in model evaluation such as "right results for wrong reasons" - when a good match of scalar variables (temperature, precipitation etc.) between a model and the reference is





achieved but the *QI* for synoptic patterns alerts about the poor model performance. We believe, such using of QIs for synoptic patterns as proposed in this study would improve evaluating routines currently used for climate models and may give valuable

feedback for model developers. We emphasize readers' attention here: the evaluation of model dynamics performed using synoptic classifications should not replace but complement (!) existing evaluation routines that use scalar variables and metrics. Another application of the synoptic classes in the evaluation of climate models is the so-called "weather-pattern-based model evaluation" (Nigro et al., 2011). Surface climate model data are analysed conditionally on each class: his allows for the determination of model errors as a function of synoptic class and can highlight if certain errors that occur under some synoptic

situation and not others. Alternatively to evaluation, other applications of the two-stage classification are possible. A linkage of synoptic classes to extreme weather could be used in improving predictability of the numerical weather models as it was done by Nguyen-Le and Yamada (2019) who classified anomalous weather patterns associated with heavy rainfall in Thailand and implemented classification results into a Global Spectral Model (GSM) of the Japan Meteorological Agency improving the forecast skill with the lead time up to 3 days. However, we doubt that using synoptic classes in a form of "precursor" for

particular weather events would be the best-suited instrument for improving weather forecasts beyond 3 days lead-time.

**Acknowledgments**

Copyright "This service is based on data and products of the European Centre for Medium-Range Weather Forecasts (ECMWF)".

**Competing interests**

The authors declare that they have no conflict of interest."

**Data sources**

European Centre for Medium-range Weather Forecast (ECMWF) (2011): The ERA-Interim reanalysis dataset, Copernicus Climate Change Service (C3S) (accessed <*insert date of access here*>), available from https://www.ecmwf.int/en/forecasts/dataset/ecmwf-reanalysis-interim

*NCEP-NCAR Reanalysis 1 data provided by the NOAA PSL, Boulder, Colorado, USA, from their website at https://psl.noaa.gov*

*CMIP6 Citation:* http://cmip6cite.wdc-climate.de



## Author Contributions

Kristina Winderlich and Clementine Dalelane designed the methods. Clementine Dalelane modified the structural similarity index metric SSIM to be applicable to mixed-sign data as presented in this study. Kristina Winderlich developed the code in FORTRAN and performed classifications, prepared the manuscript with contributions from all co-authors.

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
