# Peer review of "Classification of synoptic circulation patterns with a two-stage clustering algorithm using the modified structural similarity index metric (SSIM)"

_Earth System Dynamics, 2023_

## Referee Comment (RC1)

In this paper, the authors introduce a new clustering method for the analysis of synoptic weather types over Western Europe. The paper is primarily methodological, suggesting a new approach, discussing its robustness and demonstrating its application to climate model evaluation. They make two main points:

1. That a large number (>30) of weather types is needed to properly sample the range of synoptic flows, and so also to make sure the drivers of extreme events are included.

2. That Kmeans using RMSE has deficiencies which are fixed when using SSIM and K-Medoids.

They use a two-stage clustering procedure where the SSIM is used instead of Euclidean distance to compute distances in the K-medoids algorithm, and this is coupled to a hierarchical agglomerative model which replaces the 'number of clusters' hyperparameter with a more intuitive 'maximum similarity' hyperparameter.

They compare their new method to previous approaches using synthetic data, investigate robustness to parameter changes and to temporal resampling, and then demonstrate the application to climate model evaluation, by summarising metrics based on these weather types' occurrence, persistence and transition properties into a single overall score.

For transparency, I was reviewer 1 for both rounds of revision of the previously submitted version of this manuscript.

I have some major issues with the manuscript, detailed below, and suggest major revisions.

The usefulness of the suggested approach for capturing extreme events

One of the major motivations given for the use of a large number of clusters in full field data was that this would help capture extreme events, whereas PCA based approaches with smaller numbers of clusters may not capture extremes. Unfortunately, neither side of this claim has been demonstrated. I also have some reasons to doubt the claim: looking at their figure 8, there is not much sign that the more common weather patterns are less extreme supporting than the rare patterns. Further, other work has shown that persistent regimes (i.e. common weather types) can drive cold and warm extremes.

As I suggested previously:

"I suggest that the authors more tightly focus the structure of the article around the importance of handling rare synoptic conditions and extremes in clustering approaches, showing an example situation where an impactful event was linked to a very rarely occurring circulation as motivation. I would then suggest a concrete demonstration that the EOF Kmeans with MSE approach more poorly handles rare circulations than the SSIM approach in ERA Interim…."

Even if it is the case that rare circulations are associated with rare extremes, when you compute the Jensen-Shannon divergence, you weight each class by frequency! So representation of rare flows has almost no impact on the resulting quality index.

Usefulness for climate model evaluation

The authors also emphasise the value of their method for climate model evaluation. Indeed, circulation based metrics can be very useful for such analysis. This can and has been done several different ways (although it would be easy to think otherwise reading the authors' work), with only a few regimes at one extreme as in [1], or on a gridpoint basis as in [2] at the other extreme.

However, I am seriously concerned that the method the authors suggest is not suitable for this purpose.

The author's explain that using similarity as a metric, ~37 weather patterns are needed to fully capture the diversity of European circulations. I accept this, and it is a useful perspective, and similarity is a nice way to quantify this. Exploring spatial and seasonal variations in this number of 'necessary patterns' could be an interesting dynamical study.

But, for model evaluation, the question of relevance is not how many weather patterns you need, **but how many weather types you can constrain**, given data limitations.

The authors compute error metrics for weather pattern frequency (37 elements), transition matrix (37x37 =1369 elements) and persistence probability over days 1-8 (37x8=296 elements). Simply put, using 40 years of ERA-Interim the sampling uncertainty in such fine-grained metrics are almost certainly far larger than any difference between climate models and era-interim. The fact that the inter-model variation in scores is so low reinforces this point. I believe your quality index is almost entirely noise, averaged over a few hundred variables.

I make this claim quite confidently, as I know that it is difficult to find significant differences in the frequency and persistence of models and reanalysis when only using 3-10 regimes, and 100 years of data. Of course I would be pleased to be proven wrong: if you can rigorously constrain sampling variability in model and observational statistics, and so provide upper and lower bounds on your quality index, and still get meaningful results, then the scientific contribution is strong. Otherwise, I would move away from climate model evaluation as a goal for this methodology.

Synthetic data

The synthetic data section raises some questions for me. One clear point that I found interesting is that Kmeans leads to distorted patterns (i.e. not circles as in the synthetic data). However I think the other points would be better made in ERA Interim than in the synthetic data. The synthetic data does not have multimodal structure, so there is no reason to expect any clustering algorithm to give very clear clusters: there are no clusters to identify, just 'hallucinations' of the method. In fact, you could argue that in non-structured data, a good clustering algorithm *should* give unclear structures. Also, I do not follow the claim about snowballing: the k-medoids with SSIM produces the most snowballing of all algorithms shown in figure 4.

---

## Referee Comment (RC3)

**Classification of synoptic circulation patterns with a two-stage clustering algorithm using the modified structural similarity index metric (SSIM)**

Overview

This manuscript shows the development of a two-stage clustering algorithm that combines two classical clustering algorithms, hierarchical agglomerative clustering (HAC) and k-medoids while utilizing structural similarity index metric (SSIM) over traditional distance metrics by using the 500-hPa geopotential height from ERA-Interim reanalysis data. This approach seems to offer many benefits over traditional classification methods, and I can see it being useful for a wide range of applications within atmospheric and climate science. While the manuscript is very extensive in describing the developed algorithm and showing its robustness with reanalysis data, I have a couple of general comments and several specific comments that I believe should be addressed. I believe this paper fits well into the scope of Earth System Dynamics and I would approve this work for publication pending major revisions made to address the comments attached.

General Comments

1.  While the manuscript is very detailed in explaining and testing the methodology that was developed to classify synoptic circulations the connection and application of the method to the main motivation for its development, "to extend the evaluation routine for climate simulations", is not given the same amount of detail and attention as it should. The manuscript as is should more clearly demonstrate how the method accomplishes this objective and how it adds value to the current evaluation of climate simulations that would warrant the effort required to implement it. One possible suggestion, given the length of the paper and detail provided to the actual methodology and its testing, could be to make the application of evaluating CMIP6 simulations with this algorithm as a separate manuscript where that specific application of the method can be discussed and demonstrated in a complete manner.

2.  While there is a good discussion in the introduction with respects to building synoptic classes in pervious work there was no mention of works that used approaches such as Machine Learning and AI which is becoming more popular within Earth system science as well as other fields. For example, Gervais et al. (2016) uses Self-Organizing Maps to classify Artic Air Masses from CESM-LE. I think it would be important to discuss how approaches like SOMs, Random Forrest, etc. have been used in the classification of synoptic patterns and how this new approach compares to them.

Specific Comments

LINE 20 – Why not state what the alternative reanalysis is instead of keeping it vague by just saying "alternative reanalysis"?

LINE 175 – Would this method also work if considering more than one atmospheric variable mapped on the same domain, or can it only work with the use of a single variable?

LINE 175 – For this work, one time step a day was used, is the reason for this due to computational/time constraints or are there other issues that may arise using this method with more regular time steps, such as all timesteps in ERA-Interim or even if moving to the hourly timesteps in ERA-5. If there are restrictions associated with the method and temporal/spatial resolution of data that can be used it would be good to mention them at some point.

LINE 195 – Its not clear why NCEP1 was chosen as the alternative reanalysis compared to other available reanalysis datasets. Why would the assumption "*Assuming that the alternative reanalysis captures the synoptic circulation of the reference data ERA-Interim better than any unconstrained global circulation mode*" be made? Can more be said about this decision?

LINE 198 – I am assuming all datasets are normalized with EQ. 1? Is this correct?

LINE 375 – I'm not sure this is clear, is the "final cluster" what is used as the initialization clusters, or the final result of the entire method being presented in the manuscript?

LINE 444 – When stating "well separated …from the entire data set" does this mean the clusters should be well separated from the data that is not assigned to the given cluster?

LINE 451 – Are these "similarity diagrams" what is shown in Figure 10?

LINE 461 – If it has been established that using values such as Euclidean distance does not perform well when considering things such as synoptic patters what is the value in calculating Metric 2?

LINE 585 – To clarify, there are 183 "runs" but each run is done for varying data volumes from 1 to 40 years. So, is it correct to say the method is done 183 x 40 times? Or the output of each run is just saved after each year of data is added?

LINE 660 – It is difficult to see the dashed and grey lines in Figure 10.

LINE 805 – While I understand the reasoning for showing the 5 most frequent SP-classes one of the benefits mentioned was the ability for the algorithm to preserve less frequent patterns that are more likely to be associated with extremes. I think it is important to demonstrate this ability/benefit. I would suggest maybe showing a couple of these patterns instead of just focusing on the most frequent SP-classes.

---

## Author Comment (AC1)

Below we answer comments of the Reviewer 1.
Original comments of the Reviewer 1 are listed in black, our answers - in blue colour.

The usefulness of the suggested approach for capturing extreme events
One of the major motivations given for the use of a large number of clusters in full field data was that this would help capture extreme events, whereas PCA based approaches with smaller numbers of clusters may not capture extremes. Unfortunately, neither side of this claim has been demonstrated. I also have some reasons to doubt the claim: looking at their figure 8, there is not much sign that the more common weather patterns are less extreme supporting than the rare patterns. Further, other work has shown that persistent regimes (i.e. common weather types) can drive cold and warm extremes.
As I suggested previously:
"I suggest that the authors more tightly focus the structure of the article around the importance of handling rare synoptic conditions and extremes in clustering approaches, showing an example situation where an impactful event was linked to a very rarely occurring circulation as motivation. I would then suggest a concrete demonstration that the EOF Kmeans with MSE approach more poorly handles rare circulations than the SSIM approach in ERA Interim…."

We would like to demonstrate on the example of Germany how having many classes of synoptic patterns may help to capture extreme events. Apart from the type of extreme events mentioned by Reviewer 1, which materialize through persistence of possibly not very rare circulation types, there are others that are related to rare circulation patterns. In the figure below we show each synoptic class (left plot), the fraction (of total elements in the class) that exceed the 90-percentile near surface temperature (middle plot) and the fraction that exceed the 90-percentile in total precipitation (right plot). All data – temperature and precipitation percentiles - we computed on the ERA-Interim 1979-2018 data as the zg500 used for the classification. We validated these percentile values with the corresponding percentiles of the Germanys national HYdrological RASterdata (HYRAS) data set (https://www.dwd.de/DE/leistungen/hyras/hyras.html) and found them to match regularly (we do not show the HYRAS percentile exceedances here now, but are ready to show them upon request).

We show plots for all classes as we think it is necessary to demonstrate that some classes have no "extreme" events (SP3, 5, 9, 14, 17, 37), but some others do. Rare classes (with occurrence of less than 2% in total data) SP 25, 28, 29, 30, 34 and 35 are often "hot" i.e. show exceedances of 90-percentile in temperature. Rare classes SP 19, 20, 21, 23, 24, 29, 30, 31, 32, 36 show exceedances of 90-percentile in precipitation.

Precipitation is especially "difficult" variable to evaluate in models. Dry/wet biases in models may result from bad physical parameterisations or/and from models disability to reproduce the correct synoptic pattern. Therefore, knowing that a particular synoptic pattern often goes along with strong precipitation, we can check if a model is able to reproduce this pattern or not. This knowledge would help to attribute precipitation errors to errors in models physics or dynamics. Having only fewer synoptic patterns would hamper the differentiation between the classes and not allow us to make such attribution.

[Figure]

[Figure]

[Figure]

[Figure]

Even if it is the case that rare circulations are associated with rare extremes, when you compute the Jensen-Shannon divergence, you weight each class by frequency!
So representation of rare flows has almost no impact on the resulting quality index.

We show examples of Jensen–Shannon distance values between a 40-element Gaussian-shape histogram f(x) as reference (shown in plots by the grey dashed line) and six alternative histograms:

a) histogram of equally frequent classes (uniform distribution l(x)=const)
b) "mirrored" histogram (g(x) = – f(x)+a )
c) histogram with reduced frequency of one frequent element
d) histogram with reduced frequency of one rare element
e) histogram with 6 equally distributed frequent classes (total frequency of these 6 classes kept constant)
f) histogram with 6 equally distributed rare classes (total frequency of these 6 classes kept constant)

[Figure]

From the above plot, we see that JS=0.312 between a Gauss-shape and a uniform distributions should be considered a very large distance value as the compared distributions are obviously very different. Please note, the Jensen–Shannon divergence is bounded by 1 for two histograms (using base 2 logarithm), and therefore, JS-distance is bounded by 1 as well.
Reviewer 1 argues that rare classes have no/small impact on JS-distance, referring probably to a situation as in the plot "d": an error in frequency of one rare class makes moderate contribution to the total JS-distance (as compared to the contribution of the similar error in one frequent class, plot "c"). The plots "c" and "d" may lead the observer to a conclusion that errors in rare classes are negligible in computing JS-distance. This is a misleading

conclusion. Please see plots "e" and "f" for clarification: JS-distance is higher in response to errors in multiple rare classes (plot "f") as to the errors in frequent classes (plot "e"). The relative change in frequency of the rare classes by such error is quite high (the mean distribution M used in computation of Jensen-Shannon Divergence undergoes large changes relatively to the original distributions). The situation in our evaluation framework is most similar to plots "e" and "f", where the JS-distance appears to work quite satisfactorily.

As the *Quality Indices* (computed on JS-distance) may look much of the same magnitude, we list here our comparison of models in terms of JS-distance:

| Nr | Model name | JS for individual statistics | | | | | | | Mean JS |
|---|---|---|---|---|---|---|---|---|---|
| | | HIST | $HIST_{DFJ}$ | $HIST_{MAM}$ | $HIS_{JJA}$ | $HIST_{SON}$ | TRANSIT | PERSIST | |
| - | ERAINT(ref.reanalysis) | 0,000 | 0,000 | 0,000 | 0,000 | 0,000 | 0,000 | 0,000 | 0,000 |
| - | NCEP (alt.reanalysis) | 0,013 | 0,017 | 0,020 | 0,028 | 0,021 | 0,079 | 0,062 | 0,034 |
| 1 | ACCESS-CM2 | 0,057 | 0,115 | 0,065 | 0,125 | 0,080 | 0,165 | 0,128 | 0,105 |
| 2 | AWI-ESM-1-1-LR | 0,072 | 0,097 | 0,092 | 0,126 | 0,114 | 0,170 | 0,125 | 0,114 |
| 3 | BCC-CSM2-MR | 0,061 | 0,096 | 0,085 | 0,140 | 0,111 | 0,168 | 0,122 | 0,112 |
| 4 | BCC-ESM1 | 0,067 | 0,113 | 0,106 | 0,143 | 0,104 | 0,171 | 0,124 | 0,118 |
| 5 | CanESM5 | 0,061 | 0,124 | 0,097 | 0,091 | 0,096 | 0,174 | 0,128 | 0,110 |
| 6 | CESM2 | 0,064 | 0,093 | 0,081 | 0,116 | 0,101 | 0,164 | 0,126 | 0,107 |
| 7 | CESM2-FV2 | 0,079 | 0,125 | 0,087 | 0,138 | 0,120 | 0,181 | 0,136 | 0,124 |
| 8 | CESM2-WACCM-FV2 | 0,074 | 0,118 | 0,113 | 0,151 | 0,089 | 0,174 | 0,132 | 0,122 |
| 9 | CMCC-CM2-SR5 | 0,073 | 0,111 | 0,080 | 0,161 | 0,100 | 0,176 | 0,125 | 0,118 |
| 10 | CNRM-CM6-1 | 0,059 | 0,105 | 0,081 | 0,150 | 0,088 | 0,169 | 0,128 | 0,111 |
| 11 | CNRM-ESM2-1 | 0,043 | 0,098 | 0,087 | 0,119 | 0,089 | 0,164 | 0,126 | 0,104 |
| 12 | EC-Earth3 | 0,054 | 0,091 | 0,076 | 0,137 | 0,095 | 0,164 | 0,120 | 0,105 |
| 13 | EC-Earth3-Veg | 0,068 | 0,091 | 0,081 | 0,165 | 0,085 | 0,170 | 0,117 | 0,111 |
| 14 | FGOALS-f3-L | 0,068 | 0,147 | 0,104 | 0,173 | 0,076 | 0,170 | 0,124 | 0,123 |
| 15 | FGOALS-g3 | 0,073 | 0,141 | 0,097 | 0,145 | 0,081 | 0,175 | 0,138 | 0,121 |
| 16 | GISS-E2-1-G | 0,061 | 0,127 | 0,097 | 0,178 | 0,093 | 0,171 | 0,120 | 0,121 |
| 17 | HadGEM3-GC31-LL | 0,050 | 0,108 | 0,078 | 0,107 | 0,086 | 0,161 | 0,132 | 0,103 |
| 18 | HadGEM3-GC31-MM | 0,054 | 0,090 | 0,084 | 0,116 | 0,077 | 0,163 | 0,122 | 0,101 |
| 19 | INM-CM4-8 | 0,071 | 0,106 | 0,096 | 0,170 | 0,110 | 0,182 | 0,136 | 0,124 |
| 20 | INM-CM5-0 | 0,059 | 0,089 | 0,095 | 0,121 | 0,123 | 0,166 | 0,139 | 0,113 |
| 21 | IPSL-CM6A-LR | 0,065 | 0,099 | 0,099 | 0,181 | 0,131 | 0,169 | 0,132 | 0,125 |
| 22 | IPSL-CM6A-LR-INCA | 0,056 | 0,124 | 0,094 | 0,176 | 0,131 | 0,168 | 0,136 | 0,126 |
| 23 | KACE-1-0-G | 0,051 | 0,090 | 0,081 | 0,125 | 0,079 | 0,163 | 0,130 | 0,103 |
| 24 | MIROC6 | 0,063 | 0,105 | 0,076 | 0,136 | 0,094 | 0,164 | 0,136 | 0,111 |
| 25 | MPI-ESM-1-2-HAM | 0,061 | 0,104 | 0,085 | 0,127 | 0,104 | 0,168 | 0,122 | 0,110 |
| 26 | MPI-ESM1-2-HR | 0,057 | 0,105 | 0,082 | 0,098 | 0,088 | 0,166 | 0,118 | 0,102 |
| 27 | MPI-ESM1-2-LR | 0,056 | 0,103 | 0,070 | 0,112 | 0,085 | 0,164 | 0,124 | 0,102 |
| 28 | MRI-ESM2-0 | 0,052 | 0,090 | 0,098 | 0,122 | 0,079 | 0,161 | 0,118 | 0,103 |
| 29 | NorESM2-LM | 0,077 | 0,124 | 0,134 | 0,175 | 0,126 | 0,180 | 0,142 | 0,137 |
| 30 | NorESM2-MM | 0,065 | 0,108 | 0,087 | 0,127 | 0,126 | 0,172 | 0,129 | 0,116 |
| 31 | TaiESM1 | 0,060 | 0,121 | 0,091 | 0,119 | 0,091 | 0,166 | 0,134 | 0,112 |
| 32 | UKESM1-0-LL | 0,060 | 0,073 | 0,082 | 0,139 | 0,089 | 0,161 | 0,128 | 0,105 |
| - | *MEAN (32 models)* | *0,062* | *0,107* | *0,089* | *0,138* | *0,098* | *0,169* | *0,128* | *0,113* |
| - | *STDDEV(32 models)* | *0,008* | *0,016* | *0,013* | *0,024* | *0,017* | *0,006* | *0,007* | *0,009* |

The frequency histograms of the projections show JS distances around 0.1 (TRANSIT and PERSIST around 0.15), which is of course not way off the reference but noticeable. It would be interesting to compare that to earlier model generations to appreciate the improvements.

The computation of the *Quality Index* based on the JS-distance was done for convenience for our downstream application, which admittedly has no relevance for the contents of this manuscript:

$$QI(P \parallel Q) = exp^{-a\sqrt{JS}}$$

Besides, *QI* can be scaled by the constant *a* for a better differentiation among models. If Reviewer 1 considers it appropriate, we can skip the transformation completely.

**Usefulness for climate model evaluation**

The authors also emphasise the value of their method for climate model evaluation. Indeed, circulation based metrics can be very useful for such analysis. This can and has been done several different ways (although it would be easy to think otherwise reading the authors' work), with only a few regimes at one extreme as in [1], or on a gridpoint basis as in [2] at the other extreme.

However, I am seriously concerned that the method the authors suggest is not suitable for this purpose.

The author's explain that using similarity as a metric, ~37 weather patterns are needed to fully capture the diversity of European circulations. I accept this, and it is a useful perspective, and similarity is a nice way to quantify this. Exploring spatial and seasonal variations in this number of 'necessary patterns' could be an interesting dynamical study. But, for model evaluation, the question of relevance is not how many weather patterns you need, **but how many weather types you can constrain**, given data limitations.

It is generally acknowledged that MSE/Euclidian distance is not appropriate for use in high-dimensional spaces. Therefore, the use of prior dimension reduction is usually recommended. This drawback is less pronounced for SSIM, which makes it an alternative measure, when dimension reduction is undesirable.

All 37 classes presented in Figure 8 may look "patchy" and not different enough from each other at the first glance. However, all these classes are not similar (according to our definition) to each other as each pair of them has similarity value smaller than 0.40 (the threshold chosen for the 2-step classification algorithm). Two points are important to note: 1) This class separation is done in terms of SSIM and does not have to hold in terms of MSE. It might occur that MSE cannot differentiate between two patterns, but SSIM can. 2) Each class is represented by its medoid - this makes the separation of classes sharper and the assignment of samples less ambiguous as compared to the common practice of using centroids. The attribution of each data element to a class is done using SSIM with respect to the medoids.

The authors compute error metrics for weather pattern frequency (37 elements), transition matrix (37x37 =1369 elements) and persistence probability over days 1-8 (37x8=296 elements). Simply put, using 40 years of ERA-Interim the sampling uncertainty in such fine-grained metrics are almost certainly far larger than any difference between climate models and era-interim. The fact that the inter-model variation in scores is so low reinforces this

point. I believe your quality index is almost entirely noise, averaged over a few hundred variables.

I make this claim quite confidently, as I know that it is difficult to find significant differences in the frequency and persistence of models and reanalysis when only using 3-10 regimes, and 100 years of data. Of course I would be pleased to be proven wrong: if you can rigorously constrain sampling variability in model and observational statistics, and so provide upper and lower bounds on your quality index, and still get meaningful results, then the scientific contribution is strong. Otherwise, I would move away from climate model evaluation as a goal for this methodology.

As it is conceptually difficult to assess sampling uncertainty with only one realisation (which is furthermore not a simple Multinomial distribution), we use resampled data (10-fold block-cross validation, i.e. a sliding block of 10 years cut from the sample in a cyclic way) as to build 40 different sets of 30 year-histograms HIST, HIST_JFD, HIST_MAM, HIST_JJA, HIST_SON, and TRANSIT, PERSIST-matrices. From these 40 histograms we estimate standard deviations for each element (frequency/persistence of a certain class, transition probability from one class to another) of these one- and two-dimensional histograms.

As a very rough, zeroth-order check of robustness we compare the estimated values in the frequency histograms and the TRANSIT/PERSIST matrices with two times their resampling standard deviation. It appears that the uncertainty in the histograms is reasonably low: all values in the histogram are greater than their individual 2*std, even for the rare classes. This is in line with our earlier observation that the clustering algorithm stabilizes at around 20 years of daily data.

[Figure]

Furthermore, the values produced by the CMIP6 projections, though some do fall inside our "confidence intervals", show far higher departures from the reference than the resampled data, both in frequent and rare classes. Interestingly, many models seem to shift weight from a number of rare classes to the frequent ones, which would indicate a reduced diversity of circulation types (see figure below). In our judgment, this histogram corroborates the usefulness of our approach comparing the frequency histograms of the projections to the reference.

[Figure]

TRANSIT and PERSIST matrices.

As the Reviewer 1 suggested, some elements of the matrices TRANSIT and PERSIST do not satisfy our requirement of robustness. We show the TRANSIT-matrix along with its resampled standard deviation in the following figure (matrix elements highlighted with black contour lines are >= 2*std, std is computed over 40 resamples of TRANSIT matrices). Note that the scale is logarithmic here.

[Figure]

From the above figure we see that transition-elements with larger absolute value (higher frequencies of occurrence) are likely to be robustly estimated, small elements (rare transitions) are less likely so, as expected.

In contrast, the PERSIT-matrix seems to be sufficiently robust, i.e. all elements are greater than their 2*std. This might result from the lower number of elements contained in the matrix.

[Figure]

We suggest, that a larger data set will certainly help us to estimate all elements of the TRANSIT-matrices more robustly. For now, we agree that using about 14600 data elements

may be not enough to robustly estimate the sampling uncertainty for all elements in the transition matrix. Nonetheless, TRANSIT is certainly not "all noise".

Synthetic data

The synthetic data section raises some questions for me. One clear point that I found interesting is that K-means leads to distorted patterns (i.e. not circles as in the synthetic data). However, I think the other points would be better made in ERA Interim than in the synthetic data. The synthetic data does not have multimodal structure, so there is no reason to expect any clustering algorithm to give very clear clusters: there are no clusters to identify, just 'hallucinations' of the method. In fact, you could argue that in non-structured data, a good clustering algorithm *should* give unclear structures.

It is true that the synthetic data are generated randomly and have no genuine cluster structure of zg500. But all clustering algorithms that we know of, would produce clusters governed by the position of the largest anomaly in the domain and its sign. In the synthetic data example: the initializing clusters are produced by the HAC, which has no predefined number of classes but is only driven by the similarity threshold. Even in a completely random dataset clusters can be constructed because some samples are more similar to each other than others. In this respect, our clustering algorithm is neither better nor worse than any other algorithm.

We have chosen to show the performance of the clustering algorithms (with k-means/k-medoids using MSE/SSIM) for demonstrating the distortions of cluster centres that occur when using means, but cannot by construction occur with medoids. Using real data, these distortions would be less obvious.

Also, I do not follow the claim about snowballing: the k-medoids with SSIM produces the most snowballing of all algorithms shown in figure 4.

We don't see why Reviewer 1 is claiming that k-medoids with SSIM is producing the most "snowballing". The "snowballing" does not mean forming classes with many elements but classes with "vanishing structure", which is not the case in Figure 4 d. We explained this in the manuscript lines 317-325:

*We already showed (Figure 1 and Figure 3) that small MSE does not guarantee the structural similarity of compared patterns. Classes built with k-means-MSE show very little structural detail as a result of building cluster centroids over multiple class elements, whose structural similarity remained unaccounted. The danger of having such classes "with vanishing structure" is that they may serve as attractors for further elements as the clustering algorithm runs targeting at minimizing MSE only. This leads to the so-called "snowballing" effect i.e. the more elements are assigned to this class, the less structure shows its centroid, the more elements are assigned and so on. Cluster 9 (Figure 5) is a good example of such "snowball"-class: although all shown elements have comparable small MSE to the final class centre, their visual (for an 13 observer) and computed similarity (value of SSIM) differs strongly as shown for a group of the first 28 elements (out of 132) indicating a strong structural inhomogeneity of patterns contained in one class. This example demonstrates the danger of building "snowball" classes when using MSE as distance metric for data with highly structured patterns.*

In the classification with the ERA-Interim data, the Figure 14 shows the high similarity between the class medoids and their centroids (mean of all class elements). This means that these classes are not "snowballs". Although the classes may have many members, they show pronounced and similar (within the class) structural patterns.

---

## Author Comment (AC3)

Below we answer comments of the Reviewer 1.
Original comments of the Reviewer 1 are listed in black, our answers - in blue colour.

General Comments
1. While the manuscript is very detailed in explaining and testing the methodology that was developed to classify synoptic circulations the connection and application of the method to the main motivation for its development, "to extend the evaluation routine for climate simulations", is not given the same amount of detail and attention as it should. The manuscript as is should more clearly demonstrate how the method accomplishes this objective and how it adds value to the current evaluation of climate simulations that would warrant the effort required to implement it. One possible suggestion, given the length of the paper and detail provided to the actual methodology and its testing, could be to make the application of evaluating CMIP6 simulations with this algorithm as a separate manuscript where that specific application of the method can be discussed and demonstrated in a complete manner.
Our proposed method provides a quality index that is subsequently fed into a comprehensive evaluation routing for climate simulations along with a number of other quality indices. It is explicitly stated in the manuscript that we only "demonstrate an exemplary application" (Line 19) to show the applicability of our method for evaluation. We indeed plan to write a follow-up paper on the "full evaluation" of the CMIP6 models.

2. While there is a good discussion in the introduction with respects to building synoptic classes in pervious work there was no mention of works that used approaches such as Machine Learning and AI which is becoming more popular within Earth system science as well as other fields. For example, Gervais et al. (2016) uses Self-Organizing Maps to classify Artic Air Masses from CESM-LE. I think it would be important to discuss how approaches like SOMs, Random Forrest, etc. have been used in the classification of synoptic patterns and how this new approach compares to them.
We agree to this comment and will extend the introduction and the discussion with suggested approaches in the revised manuscript.

Specific Comments
LINE 20 – Why not state what the alternative reanalysis is instead of keeping it vague by just saying "alternative reanalysis"?
We chose NCEP1 as alternative reanalysis. But we believe that in our work it is not important which alternative reanalysis product exactly we take for demonstrating the „best" quality score. Any of available reanalysis products may be taken. The alternative reanalysis is only needed here to demonstrate the relative range of quality indices for CMIP6 models.

LINE 175 – Would this method also work if considering more than one atmospheric variable mapped on the same domain, or can it only work with the use of a single variable?
There is no "universally correct" recipe on how to build synoptic classes. As we mentioned in Lines 67-74, geopotential height and surface pressure are common variables used for the classification of synoptic patterns. But there is a variety of methods, which construct synoptic patterns on the basis of more than one variable (e.g. Bisolli&Dittmann 2001). The presented classification method can be extended to multiple variables by either targeting the optimization algorithm on a vector of similarity values, or defining the SSIM for vector-valued variables.

Bisolli, P. and Dittmann, E. (2001): The objective weather type classification of the German Weather Service and its possibilities of application to environmental and meteorological investigations. Meteorologische Zeitschrift, Vol. **10**, No. 4, 253-260

LINE 175 – For this work, one time step a day was used, is the reason for this due to computational/time constraints or are there other issues that may arise using this method with more regular time steps, such as all timesteps in ERA-Interim or even if moving to the hourly timesteps in ERA-5. If there are restrictions associated with the method and temporal/spatial resolution of data that can be used it would be good to mention them at some point.
Weather patterns are typically defined once per day sampled at 12 UTC to capture the mid-day peak in extreme weather conditions. Using more frequent output, for example 1-hourly, would increase the data volume but not qualitatively add more information on synoptic patterns as these patterns extend over scales of 1000 km and persist up to 20 days, they do not replace one another within few hours, so there is no necessity to use 1h, 3h or 6h data for classification.

LINE 195 – Its not clear why NCEP1 was chosen as the alternative reanalysis compared to other available reanalysis datasets. Why would the assumption "*Assuming that the alternative reanalysis captures the synoptic circulation of the reference data ERA-Interim better than any unconstrained global circulation mode*" be made? Can more be said about this decision?
The NCEP1 was chosen as an alternative reanalysis that covered the same period 1979-2018 as ERA-Interim, but use different models and data-assimilation routines. Any other reanalysis could be used instead. The assumption that an alternative reanalysis captures the synoptic patterns of ERA-Interim better that an unconstrained model is based on the construction of the reanalysis product: reanalysis data are updated weather forecasts initiated with the blend of past weather forecasts and the observations. Reanalysis data provide a complete and consistent picture of past weather and climate. Two different reanalysis data sets ERA-Interim and NCEP1, both assimilating real observations, can be seen as two "realizations" of the real weather/climate. Whereas a free-running (no data assimilation used) climate model can be seen as an approximation of the real weather/climate as it lacks usage of true observations. Therefore, it is natural to expect that any pair of reanalysis products that share the same observations used in their production match more closely than any pair of one reanalysis and one free-running model or even a pair of two free-running models.

LINE 198 – I am assuming all datasets are normalized with EQ. 1? Is this correct?
Yes. In Equation 1 the normalization around the 0-mean and by the standard deviation is used. It is necessary because the variance of the geopotential changes seasonally (larger in summer, smaller in winter). The normalization is done in order to be able to cluster summer and winter synoptic patterns without being over-sensitive to the higher summer variance in these fields.

LINE 375 – I'm not sure this is clear, is the "final cluster" what is used as the initialization clusters, or the final result of the entire method being presented in the manuscript?
The second. The two-stage algorithm stops, when no similar clusters are left to combine. [This is the final set of clusters.] The centres (medoids) of final clusters give the set of classes.

LINE 444 – When stating "well separated …from the entire data set" does this mean the clusters should be well separated from the data that is not assigned to the given cluster?
Cluster separation is a measure that quantifies the similarity of clusters as compared to homogeneous/random data or other clusters. We use 1) explained variation EV, 2) Euclidean distance ratio DRATIO and 3) similarity ratio SSIMRATIO. These measures characterise how clusters differ to other clusters (DRATIO and SSIMRATIO) and to the whole data set (EV).

LINE 451 – Are these "similarity diagrams" what is shown in Figure 10?
Yes. Similarity between classes derived with different merging threshold.

LINE 461 – If it has been established that using values such as Euclidean distance does not perform well when considering things such as synoptic patters what is the value in calculating Metric 2?

The metric EV is widely used to describe the separation and representability of classes in the wide community of classification methods for synoptic patterns. This metric is recommended within the project COST Action 733 report (Tveito et al., 2016) as we referred in Line 453, therefore we use it. Values of EV show that despite using medoids for building clusters, the final classes still explain a large portion of variance (although Euclidean distance was not targeted by the optimization!).

LINE 585 – To clarify, there are 183 "runs" but each run is done for varying data volumes from 1 to 40 years. So, is it correct to say the method is done 183 x 40 times? Or the output of each run is just saved after each year of data is added?

In total 183x40 runs: for each of 40 data volumes and for each of three merging thresholds 60+1 runs.

LINE 660 – It is difficult to see the dashed and grey lines in Figure 10.

We agree. We will update the Figure to make the lines more visible.

LINE 805 – While I understand the reasoning for showing the 5 most frequent SP-classes one of the benefits mentioned was the ability for the algorithm to preserve less frequent patterns that are more likely to be associated with extremes. I think it is important to demonstrate this ability/benefit. I would suggest maybe showing a couple of these patterns instead of just focusing on the most frequent SP-classes.

We agree. We will add a Figure with rare classes and corresponding extreme weather indicators to the next version of the manuscript. The maps for all SP-classes are also included in the response to Reviewer 1 (https://esd.copernicus.org/preprints/esd-2023-34/esd-2023-34-AC1-supplement.pdf).

---

## Author Response (AR1)

**Answers to comments of all Reviewers.**

Original comments of the Reviewers are listed in black, our answers - in blue colour, citations from the updated manuscript – *in blue colour and italic*.

Pages 1-6: Answers to Reviewer 1
Page 7: Answers to Reviewer 2
Pages 8-13: Answers to Reviewer 3

Answers to comments of the **Reviewer 1** with point-by-point references to modifications in the manuscript.

The usefulness of the suggested approach for capturing extreme events
One of the major motivations given for the use of a large number of clusters in full field data was that this would help capture extreme events, whereas PCA based approaches with smaller numbers of clusters may not capture extremes. Unfortunately, neither side of this claim has been demonstrated. I also have some reasons to doubt the claim: looking at their figure 8, there is not much sign that the more common weather patterns are less extreme supporting than the rare patterns. Further, other work has shown that persistent regimes (i.e. common weather types) can drive cold and warm extremes.
As I suggested previously:
"I suggest that the authors more tightly focus the structure of the article around the importance of handling rare synoptic conditions and extremes in clustering approaches, showing an example situation where an impactful event was linked to a very rarely occurring circulation as motivation. I would then suggest a concrete demonstration that the EOF Kmeans with MSE approach more poorly handles rare circulations than the SSIM approach in ERA Interim…."

We added to the manuscript a new chapter that presents potential weather extremes associated with the synoptic classes (Lines 809-832):

*5 Weather extremes affiliated with the synoptic classes*
*We compute maps of exceedance probabilities for two variables - daily near-surface air temperature tas and daily total precipitation pr – for each synoptic class using maps of exceedance of 90th-percentile for days in corresponding clusters. The computed for each class map of exceedance probability is limited to the area of Germany only as we were able to validate these data using data-sources of national observations. Figure 14 shows the maps of exceedance probabilities of 90th-percentile for temperature and precipitation affiliated with four exemplary synoptic classes. The class SP5, not a very rare one with occurrence of 3.7% in the data, has no indication to exceptionally warm or wet weather as both maps of exceedance probability remain "empty" (no exceedance). For the class SP2 the map of exceedance probability for precipitation shows a frequent exceedance of 90th-percentile everywhere in Germany with a higher probability in the southern region. The class SP35, one of the rare classes with only 0.5% of data, appears to be frequently "hot". The class SP29, also a rare one, frequently exhibits warm and wet weather conditions.*

[Figure]

*Figure 14: Examples of synoptic classes and corresponding maps of exceedance probability for temperature (tas) and precipitation (pr).*

We add new Figures S3-S7 (in supplementary) that show probability of exceedance of the 90th-percentile for temperature and precipitation for each synoptic class.

We add the description of additional data of temperature and precipitation (lines 207-212):
*Additionally to zg, we retrieve ERA-Interim daily near-surface atmosphere temperature (tas) and daily total precipitation (pr) for demonstrating potential weather extremes affiliated with each synoptic class (See Chapter 5). For these daily variables we compute 90th-percentile map on the original spatial resolution within the chosen domain over the period 1979-2018. For each daily variable we create a map of exceedance: locations where the variable exceeds its 90th-percentile gets the value of 1, otherwise – 0. These binary maps are summed up for days of the same synoptic class and normalized by the number of days in this class. Final map represents the exceedance probability for the synoptic class.*

We add to the conclusions (890-893):
*We apply the new method on the reanalysis data ERA-Interim and built a set of synoptic classes (application of the classification method on other data sets may build other sets of synoptic classes). We demonstrate that separating rare classes may be useful for diagnostics of extreme weather events affiliated with these classes. Here we clearly make use of multiple synoptic classes as only few of them would hamper such attribution.*

Even if it is the case that rare circulations are associated with rare extremes, when you compute the Jensen-Shannon divergence, you weight each class by frequency!
So representation of rare flows has almost no impact on the resulting quality index.

So representation of rare flows has almost no impact on the resulting quality index.
Usefulness for climate model evaluation The authors also emphasise the value of their method for climate model evaluation. Indeed, circulation based metrics can be very useful

for such analysis. This can and has been done several different ways (although it would be easy to think otherwise reading the authors' work), with only a
few regimes at one extreme as in [1], or on a gridpoint basis as in [2] at the other extreme. However, I am seriously concerned that the method the authors suggest is not suitable for this purpose.

*We added a new chapter "Sensitivity of Jensen-Shannon distance metric" to the manuscript (supplement) on the sensitivity of the Jensen-Shannon distance. In this chapter we show how JS-distance changes in response to distortion in original distributions in frequent and rare elements. We show that single error in a rare element makes a small contribution to the JS-distance, but multiple errors in rare elements provide large changes in the JS-distance.*

*We add to the manuscript the following statement (lines 582-585):*
*Such distance measure is robust against the "noise" from rare classes and as well as rare class-to-class transitions, but not insensitive to them. We show Jensen-Shannon distance metric on various pairs of distributions Figure S2 and discuss its sensitivity in supplement chapter "Sensitivity of Jensen-Shannon distance metric". We support this statement by the new chapter "Sensitivity of Jensen-Shannon distance metric" added to the supplement.*

*As the Quality Indices (computed on Jensen-Shannon-distance) may look much of the same magnitude, we decided to refrain from describing the Quality Index in the present manuscript (as it may be only relevant for future evaluation application) and focus on demonstrating the Jensen-Shannon distance for differentiating among CMIP6 models. Therefore, we adapted the Table 3: "CMIP6 Models and their Jensen-Shannon distances (JS)..." and the discussion of its content (lines 855-873):*
*The Jensen–Shannon distance (JS) is computed for the one-dimensional statistics (HIST, HISTDJF, HISTMAM, HISTJJA, HISTSON) as well as for the two-dimensional TRANSIT, PERSIST between the two probability distributions for each model and the reference. Resulting values of JS (Table 3) can be combined to suit objectives of the model evaluation, for example, seasonally separated JS(HISTDJF), JS(HISTMAM), JS(HISTJJA), JS(HISTSON) can be used in evaluating seasonal frequencies of synoptic patterns, JS(PERSIST) for evaluating of the duration of synoptic patterns. In this paper, we equally weight all JSs and compute the Mean Jensen-Shannon distance (Table 3). A Jensen-Shannon distance of 0.0 indicates the identity between the model and the reference. The benchmark for this study: the Mean Jensen-Shannon distance for the alternative reanalysis NCEP1 is 0.034 and can be viewed as the best possible JS for a model.*

[revised manuscript text omitted]

Usefulness for climate model evaluation

The authors also emphasise the value of their method for climate model evaluation. Indeed, circulation based metrics can be very useful for such analysis. This can and has been done several different ways (although it would be easy to think otherwise reading the authors' work), with only a few regimes at one extreme as in [1], or on a gridpoint basis as in [2] at the other extreme.

However, I am seriously concerned that the method the authors suggest is not suitable for this purpose.

The author's explain that using similarity as a metric, ~37 weather patterns are needed to fully capture the diversity of European circulations. I accept this, and it is a useful

perspective, and similarity is a nice way to quantify this. Exploring spatial and seasonal variations in this number of 'necessary patterns' could be an interesting dynamical study. But, for model evaluation, the question of relevance is not how many weather patterns you need, **but how many weather types you can constrain**, given data limitations.

*We add the following text to the manuscript (lines 614-618):*
*At the first glance at Figure 7 all 37 classes may look "patchy" and not different enough from each other. However, all these classes are not similar according to our definition as each pair of them has a similarity value smaller than 0.40 (the threshold chosen for the classification algorithm). It is important to note that as the class separation is done in terms of SSIM these classes do not have to be differentiated in terms of MSE. We showed previously (Figure 1 and Figure 3) examples of pairs of patterns that are similar in terms of MSE, but differ in terms of SSIM.*
*Also we add (lines 844-846): As each class is represented by its medoid, the class separation is sharper and the assignment of data samples less ambiguous as compared to the common practices of using centroids. The attribution of each data element to a class is done using SSIM with respect to the class medoids.*

The authors compute error metrics for weather pattern frequency (37 elements), transition matrix (37x37 =1369 elements) and persistence probability over days 1-8 (37x8=296 elements). Simply put, using 40 years of ERA-Interim the sampling uncertainty in such fine-grained metrics are almost certainly far larger than any difference between climate models and era-interim. The fact that the inter-model variation in scores is so low reinforces this point. I believe your quality index is almost entirely noise, averaged over a few hundred variables.
I make this claim quite confidently, as I know that it is difficult to find significant differences in the frequency and persistence of models and reanalysis when only using 3-10 regimes, and 100 years of data. Of course I would be pleased to be proven wrong: if you can rigorously constrain sampling variability in model and observational statistics, and so provide upper and lower bounds on your quality index, and still get meaningful results, then the scientific contribution is strong. Otherwise, I would move away from climate model evaluation as a goal for this methodology.

*We complemented the manuscript with additional part "Analysis of the robustness of the estimates for the statistics HIST/TRANSIT/PERSIST" (in supplement), where we analyse the robustness of the estimated statistics. We added to the manuscript the following part (lines 838-844):*
*As we suggest using the statistics HIST, TRANSIT and PERSIST for evaluation of climate models, a question on the robustness of these statistics may arise. We take 40 sub-samples (30 years each) of the original ERA-Interim full data set of 1979-2018 and assign these data to the final 37 synoptic classes. For each statistic we compute the mean and the standard deviation (sd) of these 40 re-samples. As a very rough, zeroth-order check of robustness we compare the estimated values in the frequency histograms and the TRANSIT/PERSIST matrices with two times their resampling standard deviation. We discuss the results of this analysis of robustness in detail in the supplement ("Analysis of the robustness of the estimates for the statistics HIST/TRANSIT/PERSIST").*

We added the chapter "Analysis of the robustness of the estimates for the statistics HIST/TRANSIT/PERSIST" to the supplement.

Synthetic data
The synthetic data section raises some questions for me. One clear point that I found interesting is that K-means leads to distorted patterns (i.e. not circles as in the synthetic data). However, I think the other points would be better made in ERA Interim than in the synthetic data. The synthetic data does not have multimodal structure, so there is no reason to expect any clustering algorithm to give very clear clusters: there are no clusters to identify, just 'hallucinations' of the method. In fact, you could argue that in non-structured data, a good clustering algorithm *should* give unclear structures.

We have re-written the part of the text to make it more clear (Lines 166-172):
*We generated this synthetic data using Gaussian-shaped anomalies trying to mimic the smooth shape of geopotential patterns (the real data we wish to use later) and to illustrate how such anomalies are treated by the classification algorithm. The synthetic data are generated randomly and have no genuine structure of the geopotential patterns. However, any clustering algorithm should produce clusters governed by the position of the largest anomaly in the domain and its sign. The original circular shapes of the synthetic generated data help to illustrate how such shapes are grouped into classes by classifications in a simpler manner as if we would have used the real data for this demonstration (using real data makes these distortions less obvious).*

Also, I do not follow the claim about snowballing: the k-medoids with SSIM produces the most snowballing of all algorithms shown in figure 4.

The "snowballing" does not mean forming classes with many elements but classes with "vanishing structure" (i.e. with dissimilar elements in one class), which is not the case in Figure 4 d. We explained this in the manuscript lines 317-325:
*We already showed (Figure 1 and Figure 3) that small MSE does not guarantee the structural similarity of compared patterns. Classes built with k-means-MSE show very little structural detail as a result of building cluster centroids over multiple class elements, whose structural similarity remained unaccounted. The danger of having such classes "with vanishing structure" is that they may serve as attractors for further elements as the clustering algorithm runs targeting at minimizing MSE only. This leads to the so-called "snowballing" effect i.e. the more elements are assigned to this class, the less structure shows its centroid, the more elements are assigned and so on. Cluster 9 (Figure S1) is a good example of such "snowball"-class: although all shown elements have comparable small MSE to the final class centre, their visual (for an observer) and computed similarity (value of SSIM) differs strongly as shown for a group of the first 28 elements (out of 132) indicating a strong structural inhomogeneity of patterns contained in one class. This example demonstrates the danger of building "snowball" classes when using MSE as distance metric for data with highly structured patterns.*

We added also (Lines 800-803):
*Figure 13 shows the high similarity between the class medoids and their centroids and indicates that these classes are not "snowballs": although the classes may have many members, they show pronounced and similar (within the class) structural patterns.*

Answers to comments of the **Reviewer 2** with point-by-point references to modifications in the manuscript.

The authors have made a significant effort to improve their manuscript upon my earlier suggestions.
Nevertheless, I would like to read the author's response to Reviewer 1 raised weaknesses before recommending acceptance.

We hope Reviewer 2 would find our answers to comments of Reviewer 1 convincing.

Answers to comments of the **Reviewer 3** with point-by-point references to modifications in the manuscript.

General Comments
1. While the manuscript is very detailed in explaining and testing the methodology that was developed to classify synoptic circulations the connection and application of the method to the main motivation for its development, "to extend the evaluation routine for climate simulations", is not given the same amount of detail and attention as it should. The manuscript as is should more clearly demonstrate how the method accomplishes this objective and how it adds value to the current evaluation of climate simulations that would warrant the effort required to implement it. One possible suggestion, given the length of the paper and detail provided to the actual methodology and its testing, could be to make the application of evaluating CMIP6 simulations with this algorithm as a separate manuscript where that specific application of the method can be discussed and demonstrated in a complete manner.

Our proposed method provides only a metric that can be used in a comprehensive evaluation routing for climate simulations along with a number of other [already existing] metrics. We state this in the manuscript more clearly (lines 20-24):
*We demonstrate an exemplary application of the synoptic circulation classes obtained with the new classification method for evaluating CMIP6 historical climate simulations and an alternative reanalysis (for comparison purposes): output fields of CMIP6 simulations (and of the alternative reanalysis) are assigned to the classes and the Jensen-Shannon distance is computed for the match in frequency, transition and duration probabilities of these classes. We propose using this distance metric to supplement a set of commonly used metrics for model evaluation.*

We clarify the following statement in the conclusions (lines 899-905):
*Using the distance metric proposed in this study would help to avoid misinterpretations in model evaluation such as "right results for wrong reasons" - when a good match of scalar variables (temperature, precipitation etc.) between a model and the reference is achieved but the distance metric for synoptic patterns alerts about poor model performance. We believe, the use of such distance metric for synoptic patterns as proposed in this study would improve evaluating routines currently used for climate models and may give valuable feedback for model developers. We emphasize readers' attention here: the evaluation of model dynamics performed using synoptic classifications should not replace but complement (!) existing evaluation routines that use scalar variables and other metrics.*

We indeed plan to write a follow-up paper on the "full evaluation" of the CMIP6 models.

2. While there is a good discussion in the introduction with respects to building synoptic classes in pervious work there was no mention of works that used approaches such as Machine Learning and AI which is becoming more popular within Earth system science as well as other fields. For example, Gervais et al. (2016) uses Self-Organizing Maps to classify Artic Air Masses from CESM-LE. I think it would be important to discuss how approaches like SOMs, Random Forrest, etc. have been used in the classification of synoptic patterns and how this new approach compares to them.

We extended the introduction (lines 118-124):

*A relatively new group of synoptic classification methods uses self-organizing maps (Kohonen, 2001). These SOM methods employ a neural network algorithm that discovers patterns in data in an unsupervised way. Such algorithms have an advantage as compared to methods based on the principal component analysis (PCA) and subsequent clustering of data as the SOM do not require orthogonality and stationarity of identified classes. Studies that use the SOM-technique to classify synoptic patterns and relate these patterns to local weather (Cassano et al., 2006; Gervais et al., 2016; Hewitson and Crane, 2002; Jiang et al., 2011) typically use a pre-defined number of classes and employ the Euclidean distance measure for similarity between data elements and centroids for representing cluster centres.*

And added this (Line 233-235):
*Although k-means and its multiple variants, as well as the more general group of SOM-based methods with neighbour radius ≥1, are commonly applied in the field of the atmospheric science, they exhibit serious limitations with regard to our aims…*

Unfortunately, we were not able to find in [available to us] literature any application of random forests to classifications of synoptic weather patterns.

Specific Comments

LINE 20 – Why not state what the alternative reanalysis is instead of keeping it vague by just saying "alternative reanalysis"?

We added a new comment on the choice of the reanalysis data to make it clearer (Lines 216-220):
*The third data set is the alternative reanalysis NCEP1 (Kalnay et al., 1996). Any other reanalysis dataset may be taken. Assuming that the alternative reanalysis captures the synoptic circulation of the reference data ERA-Interim (both reanalysis product use and share at least some portion of global weather observations) better than any unconstrained global circulation model, the evaluation of an alternative reanalysis gives an estimate of the lower bound for the attainable value of the distance metric.*

LINE 175 – Would this method also work if considering more than one atmospheric variable mapped on the same domain, or can it only work with the use of a single variable?

We add (lines 68-75):
*Weather patterns can be defined at a regular temporal step, typically one day (Lamb, 1972; Hess and Brezowsky, 1952; Fabiano et al., 2020; Cannon, 2012) and be classified independent on their duration (James, 2006; Cannon, 2012; Beck et al., 2007; Fettweis et al., 2010). Alternatively, only recurrent, quasi-stationary and temporally persistent states of the atmospheric circulation would be classified (Dorrington and Strommen, 2020; Hochman et al., 2021) eliminating short-term patterns in the final set of classes.*
*There is no "universally correct" recipe on how to build synoptic classes and how many of them. Each application requires a number of classes constructed in a way best suitable for its purposes. A set of classes can be determined subjectively by an expert, as the well-known Hess-Brezowski Grosswetterlagen (Gerstengarbe and Werner, 1993; James, 2006; Hess and Brezowsky, 1952) or the Lamb weather types (Lamb, 1972), or using an automated classification method.*

*Weather situations are often described as patterns of positive and negative anomalies of geopotential (Hochman et al., 2021; Fabiano et al., 2020; Fettweis et al., 2010) or surface pressure (Lund, 1963; Beck et al., 2007), or a combination of both (Cannon, 2012; James, 2006) seen together at the horizontal scale of about 1000 km (synoptic scale).*

But there is a variety of methods, which construct synoptic patterns on the basis of more than one variable (e.g. Bisolli&Dittmann 2001).

- Bisolli, P. and Dittmann, E. (2001): The objective weather type classification of the German Weather Service and its possibilities of application to environmental and meteorological investigations. Meteorologische Zeitschrift, Vol. 10, No. 4, 253-260

We also add a comment on the possible extension of the method (lines 886-889):
*In this paper we describe the method – the recipe – to build a set of synoptic classes. We do not propose an "optimal classification" of synoptic patterns for all purposes. Depending on the purpose of classification, the presented classification method can be extended (from the single variable - geopotential anomaly at 500 hPa) to multiple variables by either targeting the optimization algorithm on a vector of similarity values, or defining the SSIM for vector-valued variables.*

LINE 175 – For this work, one time step a day was used, is the reason for this due to computational/time constraints or are there other issues that may arise using this method with more regular time steps, such as all timesteps in ERA-Interim or even if moving to the hourly timesteps in ERA-5. If there are restrictions associated with the method and temporal/spatial resolution of data that can be used it would be good to mention them at some point.

We added a more detailed description of the data selection to the manuscript (lines 182-187):
*Simulated synoptic regimes are represented by the geopotential height (zg) at the pressure level of 500hPa sampled daily at 12:00 UTC for two practical reasons: 1) it often matches the mid-day peak in extreme weather conditions and 2) it is a typically available time for model output (for subsequent model evaluation). There is no necessity of using more frequent fields, for example 1-, 3-, or 6-hourly, as this would increase the data volume but would not add more information on the synoptic patterns: these patterns do not replace each other in few hours but extend over large spatial scales and may persist for several days or longer.*

LINE 195 – Its not clear why NCEP1 was chosen as the alternative reanalysis compared to other available reanalysis datasets. Why would the assumption "Assuming that the alternative reanalysis captures the synoptic circulation of the reference data ERA-Interim better than any unconstrained global circulation mode" be made? Can more be said about this decision?

We refer to our comment on the choice of the reanalysis data (Lines 216-220) as above:
*The third data set is the alternative reanalysis NCEP1 (Kalnay et al., 1996). Any other reanalysis dataset may be taken. Assuming that the alternative reanalysis captures the synoptic circulation of the reference data ERA-Interim (both reanalysis product use and share at least some portion of global weather observations) better than any unconstrained global*

*circulation model, the evaluation of an alternative reanalysis gives an estimate of the lower bound for the attainable value of the distance metric.*

The assumption that an alternative reanalysis captures the synoptic patterns of ERA-Interim better that an unconstrained model is based on the construction of the reanalysis product: reanalysis data are updated weather forecasts initiated with the blend of past weather forecasts and the observations. Two different reanalysis data sets ERA-Interim and NCEP1, both assimilating real observations, can be seen as two "realizations" of the real weather/climate.

LINE 198 – I am assuming all datasets are normalized with EQ. 1? Is this correct?

Yes. In Equation 1 the normalization around the 0-mean and by the standard deviation is used. It is necessary because the variance of the geopotential changes seasonally (larger in summer, smaller in winter). The normalization is done in order to be able to cluster summer and winter synoptic patterns without being over-sensitive to the higher summer variance in these fields.
We added the following explanation to the manuscript (lines 195-197):
*Some typical synoptic patterns may occur in different seasons but should be grouped into one class. As the mean and the variance of the geopotential change seasonally (larger in summer, smaller in winter) the original data should be pre-processed in order to reduce the sensitivity of the classification to the summer variance and the mean in the data.*

LINE 375 – I'm not sure this is clear, is the "final cluster" what is used as the initialization clusters, or the final result of the entire method being presented in the manuscript?

The second. The two-stage algorithm stops, when no similar clusters are left to combine. [This is the final set of clusters.] The centres (medoids) of final clusters give the set of classes. We specified this now clearly in the text (lines 388-390):
*Some typical synoptic patterns may occur in different seasons but should be grouped into one class. As the mean and the variance of the geopotential change seasonally (larger in summer, smaller in winter) the original data should be pre-processed in order to reduce the sensitivity of the classification to the summer variance and the mean in the data.*

LINE 444 – When stating "well separated …from the entire data set" does this mean the clusters should be well separated from the data that is not assigned to the given cluster?

Cluster separation is a measure that quantifies the similarity of elements within clusters as compared to homogeneous/random data or other clusters. We use 1) explained variation EV, 2) Euclidean distance ratio DRATIO and 3) similarity ratio SSIMRATIO measures to quantify this separation. These measures characterise how clusters differ to other clusters (DRATIO and SSIMRATIO) and to the whole data set (EV). Ideally the EV must be as close as possible to 1, DRATIO – as small as possible (close to 0 ) and SSIMRATIO as high as possible. This follows from the definitions of these metrics.

LINE 451 – Are these "similarity diagrams" what is shown in Figure 10?

Yes. Similarity between classes derived with different merging threshold.

LINE 461 – If it has been established that using values such as Euclidean distance does not perform well when considering things such as synoptic patters what is the value in calculating Metric 2?

The metric EV is widely used to describe the separation and representability of classes in the wide community of classification methods for synoptic patterns. This metric is recommended within the project COST Action 733 report (Tveito et al., 2016) as we referred in Line 463. We use it to show how it degrades with tightening the threshold on similarity for cluster building. Values of EV show that - despite using medoids for building clusters - the final classes still explain a large portion of variance (although Euclidean distance was not targeted by the optimization!). We say (lines 670-671):
*Please note: metrics EV and DRATIO illustrate only (!) the influence of the $TH_{merge}$ on the final set of classes and do not describe the quality of classes as they are computed using the Euclidean Distance – a measure that was not optimized by the clustering algorithm.*

LINE 585 – To clarify, there are 183 "runs" but each run is done for varying data volumes from 1 to 40 years. So, is it correct to say the method is done 183 x 40 times? Or the output of each run is just saved after each year of data is added?

In total 183x40 runs: for each of 40 data volumes and for each of three merging thresholds 60+1 runs.

LINE 660 – It is difficult to see the dashed and grey lines in Figure 10.

We agree. We updated this figure. It is Figure 9 now (its number changed as some figures were moved to the supplement in the present version of the manuscript).

LINE 805 – While I understand the reasoning for showing the 5 most frequent SP-classes one of the benefits mentioned was the ability for the algorithm to preserve less frequent patterns that are more likely to be associated with extremes. I think it is important to demonstrate this ability/benefit. I would suggest maybe showing a couple of these patterns instead of just focusing on the most frequent SP-classes.

We agree. We added a new chapter to the manuscript (Lines 809-832):

*5 Weather extremes affiliated with the synoptic classes*

*We compute maps of exceedance probabilities for two variables - daily near-surface air temperature tas and daily total precipitation pr – for each synoptic class using maps of exceedance of 90th-percentile for days in corresponding clusters. The computed for each class map of exceedance probability is limited to the area of Germany only as we were able to validate these data using data-sources of national observations. Figure 14 shows the maps of exceedance probabilities of 90th-percentile for temperature and precipitation affiliated with four exemplary synoptic classes. The class SP5, not a very rare one with occurrence of 3.7% in the data, has no indication to exceptionally warm or wet weather as both maps of exceedance probability remain "empty" (no exceedance). For the class SP2 the map of exceedance probability for precipitation shows a frequent exceedance of 90th-percentile everywhere in Germany with a higher probability in the southern region. The class SP35, one*

*of the rare classes with only 0.5% of data, appears to be frequently "hot". The class SP29, also a rare one, frequently exhibits warm and wet weather conditions.*

[Figure]

*Figure 14: Examples of synoptic classes and corresponding maps of exceedance probability for temperature (tas) and precipitation (pr).*

We add new Figures S3-S7 (in supplementary) that show probability of exceedance of the 90th-percentile for temperature and precipitation in each synoptic class.

---

## Referee Report (RR1)

Review of "Classification of synoptic circulation patterns with a two-stage clustering algorithm using the modified structural similarity index metric (SSIM)"

In my previous review I raised some major issues: The claim that a large number of full field regimes would help capture extremes was unproven. The suitability of the JS index for capturing extreme event errors was unclear. The skill metrics used were very granular and it was not obvious to me they could be usefully constrained in observational or simulation data.

All these points have now been addressed, however I am not fully convinced by the treatment of the third point.
Taking the paper's results as given, the sensitivity testing of the HIST metrics and others to resampling look very good: it is easy to see that the model errors are far outside the 2*std range shown in many cases.

However, I was really surprised to see such small sampling errors. Taking one exemplary case: if we look at class 37 in figure S8, the mean occurrence is around 0.3%, with a standard deviation clearly much lower than 0.1%! I believe these small stds are down to the way the bootstrap is computed (3 ten-year samples), and give a spurious impression of how well the elements are constrained.

To investigate, I coded up a synthetic example, randomly sampling from 37 states with true occurrence frequency:
([0.003,0.003,0.005,0.005,0.008,0.008,0.01,0.01,
 0.0275,0.0275,0.0275,0.0275,0.0275,0.0275,0.0275,0.0275,
 0.0275,0.0275,0.0275,0.0275,0.0275,0.0275,0.0275,0.0275,
 0.0275,0.0275,0.0275,0.0285,0.0285,0.0295,0.0295,0.0295,
    0.03,0.03,0.03,0.05,0.14])

I took a 39*365 day sample as my 'ERA INTERIM truth', and then computed sampling errors using the method described in the new manuscript. I get a result very similar (qualitatively) to the authors, plotting error bars as 2*std as they do:

[Figure]

However, if I produce 1000 39-year samples of my state vector, and assess the actual sampling error (i.e. the expected error between two 39 year sampling periods or ERA-Interim and an uninitialised climate model run) I get the following plot:

[Figure]

The bootstrap is clearly massively underestimating the sampling error. Of course the authors can not magically obtain another 1000 years of ERA Interim data, but the point is that sampling 30 years from 39 years does not provide sufficiently independent samples to well represent errors.

**Dropping the bootstrap down to 20 years (2 blocks of 10) gives much more reasonable results and I recommend the authors do this** and then reanalyse their conclusions. Alternatively they could download ERA5 data and repeat the analysis, as that now covers a 73 year period, and allows for much larger independent bootstraps.

[Figure]

As I consider the current error quantification to be (accidentally) misleading, I am again recommending major revisions. All other aspects of the manuscript are much improved however.

---

## Author Response (AR2)

**Answers to comments of all Reviewers.**
Original comments of the Reviewers are listed in black, our answers - in blue colour.

Answer to Reviewer 1.Firstly, we would like to thank reviewer 1 for going to such lengths and calculate his own simulations to illustrate his issue. Our comments on these simulations are the following: As already noted in our previous answer, the distribution of the synoptic patterns within the sample of 40 years of ERAINT is not a simple multinomial. Considering a single day a random variable, which can take one of 37 states (one state=one synoptic class), the days are neither independent nor identically distributed. Instead, the succession of weather patterns is highly auto-correlated (a great number of patterns is simply not possible to succeed each other) and the distribution depends heavily on the season and on low frequency climate variability. These preconditions narrow down the state space of the sequence considerably. Unfortunately, we are not able to quantify this restriction. But it entails the inconvenient consequence that the simple multinomial suggested by reviewer 1 is largely inappropriate as a null model for our data. A large portion of the state space spanned by the multinomial distribution, as far as physically possible, would generate a climate that differs from historic climate. What portion of the state space exactly would generate a different climate, how much different this climate would be and by which measures, we don't know. But the confidence intervals presented by reviewer 1 in his 2nd figure are definitely not applicable in our case.
As suggested by reviewer 1, we have repeated our resampling analysis with 20 instead of 30 years for each bootstrap sample. The results look very similar (to the Figure S9 in the supplement of our manuscript), albeit with wider confidence intervals as expected (see figure below):

[Figure]

This Figure (above) does not resemble the 3rd figure of reviewer 1. But as shown in Figure 6 of our manuscript, 20 years of daily data is the very lower limit of data necessary to capture all synoptic patterns that occur in the historical period. We therefore advised for longer data periods and do not consider the 20-year resampling a proper analysis of sampling error.

We believe that the sampling error is quite appropriately captured in the analysis of the alternative reanalysis NCEP1, which was generated by a completely different model in a completely independent attempt to reconstruct the historic climate, but of course based on the same observations ensuring that the state space is equal. The histogram of synoptic patterns captured in this alternative reanalysis NCEP1 is shown in the figure below (red colour):

[Figure]

The frequencies of each synoptic pattern in ERAINT and NCEP1 are close to each other. The NCEP1 frequencies lie within a 2*sdev interval of ERAINT for all synoptic patterns except one (SP class 35, 3rd from left in the above figure), which is nevertheless very close to its upper bound. In fact, one out of 37 classes to overshoot the *2\*stdev* by a little margin is exactly what should be expected. The closeness of the NCEP1-genrated histogram to the reference histogram gives us an additional evidence that the frequencies of synoptic patterns are estimated quite well.

We add this latter Figure (with NCEP1-frequencies) and our discussion to the supplement of the manuscript. We hope this strengthen our argument about robustness of estimates for the SP-class frequencies.

Answer to Reviewer 2. Reviewer 2 recommended to accept the Manuscript without corrections.
Answer to Reviewer 3.
INE 869 – Should this line say, "whereas other models have *lower* values" not "higher values"?
Yes, we agree. This is a typo we overlooked to change as we switch from the Quality Index (best value is 1.0) to the Jensen-Shannon distance (best value is 0.0). In terms of HS-distance, the better performance of a Model is shown by a "lower" value as reviewer 3 noticed. Thank you very much!
We corrected the text to the following (Lines 868-869): *"… the climate simulation NorESM2-LM seems to underperform all other models (Mean JS=0.137) whereas other models show lower values (i.e. smaller distances to the reference statistics)."*